# A conserved chronobiological complex times *C. elegans* development

Rebecca K Spangler [1,2,7], Kathrin Braun [3,7], Guinevere E Ashley [4,7], Marit van der Does [3,5], Daniel Wruck[1], Andrea Ramos Coronado[1], James Matthew Ragle [4], Vytautas Iesmantavicius[3], Lucas J Morales Moya[3], Keya Daly[4], Carrie L Partch [1,2,6], Helge Großhans [3,5] & Jordan D Ward [4]

## Abstract

The mammalian protein PERIOD (PER) and its *C. elegans* orthologue LIN-42 have been proposed to constitute an evolutionary link between two distinct, circadian and developmental, timing systems. While the function of PER in animal circadian rhythms is well understood molecularly and mechanistically, this is not true for LIN-42's function in timing rhythmic development, reflected in *C. elegans* molting cycles. We observed arrhythmic molts upon combined deletion of a region comprising two distinct sequence elements previously termed SYQ and LT. This region functions as a casein kinase I (CK1)-binding domain (CK1BD) mediating stable binding to KIN-20, the *C. elegans* CK1δ/ε orthologue. CK1 phosphorylates LIN-42, and the CK1BD sub-domains SYQ/CKBD-A and LT/CKBD-B play distinct roles in controlling CK1-binding and kinase activity in vitro. KIN-20 and the LIN-42 CK1BD are required for proper molt timing in vivo, and loss of LIN-42 binding or of the phosphorylated LIN-42 tail impairs nuclear accumulation of KIN-20. These findings indicate that LIN-42/PER and KIN-20/CK1 form a functionally conserved signaling module of two distinct chronobiological systems.

**Keywords** Developmental Clock; Period/LIN-42; Casein Kinase I/KIN-20; Chronobiology; *C. elegans*
**Subject Categories** Development; Signal Transduction

See also: KY Hui & JA Ripperger

## Introduction

Chronobiology is the study of biological timekeeping. Circadian rhythms that enable organisms to anticipate daily cycles of light, temperature, and other environmental variables are one of the best studied examples (Takahashi, 2017). Mammalian PERIOD (PER) proteins are central to this timekeeping process and their cellular abundance, stability, and nuclear accumulation changes rhythmically over a ~24-h cycle to dictate circadian period. In mammals, the paralogs PERIOD1 and PERIOD2 (PER1 and PER2) associate with Casein kinase 1 δ/ε (CK1δ/ε) as well as the CRYPTO-CHROME proteins, CRY1 and CRY2, to form nuclear complexes that repress the transcriptional activators circadian locomotor output cycles protein kaput (CLOCK) and brain and muscle ARNT-like 1 (BMAL1) (Appendix Fig. S1A) (Gallego and Virshup, 2007; Gekakis et al, 1998; Lee et al, 2001; Lowrey and Takahashi, 2011). Since the targets of the CLOCK-BMAL1 heterodimer include PER and CRY themselves, PER and CRY thus eventually repress their own production, permitting subsequent re-expression of CLOCK-BMAL1 to start a new cycle (Gallego and Virshup, 2007; Preußner and Heyd, 2016).

To maintain the 24-h period of circadian rhythms, a delay between the activation and the repression of CLOCK-BMAL1 transcription is required. Central to this delay is the post-translational modification of PER by one of the two closely related CK1δ and CK1ε kinases. For convenience and because of the redundancy in their function, we will henceforth refer to the two isoforms generically as CK1. Anchored through a CK1-Binding Domain (CK1BD) (Lee et al, 2004), CK1 is associated with PER for its entire existence in the cell, even translocating from the cytoplasm to the nucleus together (Appendix Fig. S1A)(Aryal et al, 2017; Lee et al, 2001), allowing PER to deliver CK1 to other targets at clock-controlled promoters (Aryal et al, 2017; Cao et al, 2021). CK1-mediated phosphorylation of a PER degron licenses PER ubiquitylation and subsequent degradation, thereby limiting PER abundance and its period of activity (Appendix Fig. S1B) (Akashi et al, 2002; Eide et al, 2005; Shirogane et al, 2005). In addition to the degron, CK1 phosphorylates multiple serines in another PER sequence element known as FASP ("Familial Advanced Sleep Phase") (Appendix Fig. S1B) (Narasimamurthy et al, 2018; Philpott et al, 2023). Phosphorylation of the FASP site stabilizes PER and turns the region into an inhibitor of CK1 kinase activity (Philpott et al, 2020; Xu et al, 2007). Mutations in PER or CK1 that disrupt the balance between degron and FASP phosphorylation, result in altered circadian period. For example, in humans, a single residue

[1]Department of Chemistry and Biochemistry, University of California-Santa Cruz, Santa Cruz, CA 95064, USA. [2]Howard Hughes Medical Institute, University of California-Santa Cruz, Santa Cruz, CA 95064, USA. [3]Friedrich Miescher Institute for Biomedical Research, Basel 4056, Switzerland. [4]Department of Molecular, Cell, and Developmental Biology, University of California-Santa Cruz, Santa Cruz, CA 95064, USA. [5]University of Basel, Basel 4002, Switzerland. [6]Center for Circadian Biology, University of California-San Diego, La Jolla, CA 92093, USA. [7]These authors contributed equally: Rebecca K Spangler, Kathrin Braun, Guinevere E Ashley. ✉E-mail: helge.grosshans@fmi.ch; jward2@ucsc.edu

mutation that blocks all FASP phosphorylation results in a short circadian period that manifests as the eponymous Familial Advanced Sleep Phase, where affected individuals wake up very early in the morning (Jones et al, 1999; Toh et al, 2001).

In contrast to circadian timing, the chronobiology of development is mechanistically less well understood. In *C. elegans*, a single PER orthologue, LIN-42, appears to diverge substantially in structure and function from the mammalian and fly PER proteins. At 598 amino acids for its longest isoform, LIN-42 is substantially shorter than mouse PER2 at 1225 amino acids (Jeon et al, 1999). LIN-42 lacks a CRY-binding domain, consistent with a lack of a CRY orthologue in *C. elegans*, and in its predicted tandem PER-ARNT-SIM (PAS) domains, only PAS-B is well conserved, adopts a canonical fold, and mediates dimerization in a mode identical to mammalian PER (Lamberti et al, 2024; Migliori et al, 2023). Finally, two stretches of sequence, previously termed SYQ and LT according to their first amino acids (Tennessen et al, 2006), bear sequence homology to the two PER CK1BD subdomains, CK1BD-A and CK1BD-B, but their function in *C. elegans* has remained unexplored. *lin-42* further differs from canonical PERs in its expression dynamics and functions: rather than exhibiting a ~24-h, temperature-invariant period, *lin-42* expression cycles exhibit a temperature-dependent length that can be as short as ~7 h at 25 °C (Gissendanner et al, 2004; Jeon et al, 1999). Indeed, *lin-42* was identified as a developmental gene whose mutation causes heterochronic phenotypes, i.e., defects in temporal cell fate specification where cells adopt adult cell fates precociously, in larvae (Abrahante et al, 1998; Liu, 1990). Hence, it was proposed that *lin-42/PER* constitute an evolutionary link between two distinct, circadian and developmental, timing systems (Jeon et al, 1999).

The extent of functional similarity between these timing systems generally, and LIN-42 and PER function specifically, has remained uncertain, despite the realization that additional orthologues of mammalian clock genes exist and cause heterochronic phenotypes when mutated (Banerjee et al, 2005; Gissendanner et al, 2004; Hasegawa et al, 2005; Jeon et al, 1999; Kostrouchova et al, 1998; Migliori et al, 2023). Conceptually, whereas mutations of circadian clock genes such as *PER* change the tempo and/or robustness of circadian rhythms, heterochronic mutations are defined by their ability to alter the sequence of developmental events, such that certain events are skipped (Ambros, 1989; Ambros and Horvitz, 1984). Accordingly, and despite its rhythmic expression, the heterochronic function of LIN-42 does not appear to involve recurring activity but rather the stage-specific repression of the *let-7* miRNA prior to the third larval stage (McCulloch and Rougvie, 2014; Perales et al, 2014; Van Wynsberghe et al, 2014).

LIN-42 is also required for the rhythmic occurrence of molting, i.e., the process in which wild-type animals regenerate a new collagenous apical extracellular matrix (cuticle), at the end of each of the four larval stages (Lažetić and Fay, 2017). Under constant environmental conditions, individually grown animals enter and exit molts with great temporal uniformity, revealing robust temporal control. *lin-42(ok2385)* mutant animals were shown to cause a slow-down of development as well as an arrhythmic molting phenotype where this uniformity is lost such that individual animals molt at different times (Monsalve et al, 2011; Olmedo et al, 2015). It is unknown how LIN-42 contributes to rhythmic molting mechanistically. In addition to *CRY*, the *C.*

*elegans* genome lacks obvious orthologues of *BMAL1* and *CLOCK*, arguing for a mechanism that differs from that of the circadian clock at least in the identity of several core components (Hasegawa et al, 2005). In a yeast two-hybrid assay, LIN-42 was shown to be capable of binding to numerous transcription factors (Kinney et al, 2023). Although the functional relevance of such binding for molting has remained unexplored, LIN-42 binding to REV-ERB, a mammalian circadian clock component, orthologue NHR-85 appears important for robust periodic transcription of the heterochronic *lin-4* miRNA. Finally, loss of KIN-20, the *C. elegans* orthologue of CK1δ/ε, replicates some heterochronic phenotypes of *lin-42* mutation and slows development (Banerjee et al, 2005), but its heterochronic functions were argued not to involve LIN-42 (Rhodehouse et al, 2018), and it is unknown whether KIN-20 is required for rhythmic molting.

Here, we set out to further characterize the molecular and developmental functions of LIN-42. Using targeted mutations, we find that the N-terminal PAS region—that includes the single bona fide PAS-B domain and an adjacent structured but unresolved element—is largely dispensable both for heterochronic pathway activity and rhythmic molting. By contrast, the SYQ/LT regions are specifically required for periodic molting. We demonstrate that these domains function as a CK1BD, mediating stable KIN-20/CK1 binding to LIN-42 in vivo and in vitro. CK1 phosphorylates LIN-42 in vitro and this activity is distinctly modulated through the two CK1BD subdomains: LT/CK1BD-B is particularly important for stable CK1 binding, whereas the SYQ/CK1BD-A element modulates product release and CK1 enzymatic activity. Moreover, loss of LIN-42 binding impairs nuclear accumulation of KIN-20. Our results identify LIN-42/PER–KIN-20/CK1 as a conserved chronobiological complex utilized by two distinct biological oscillators. Although CK1 activity in the circadian clock has previously been viewed mostly through the lens of its effects on PER, our findings align well with the growing notion that PER-mediated regulation of CK1 may be an additional important mechanism to support robust circadian rhythms.

## Results

### The LIN-42 SYQ-LT motifs mediate a subset of developmental functions of LIN-42

*lin-42(null)* animals are viable yet exhibit highly penetrant heterochronic and molting phenotypes (Edelman et al, 2016). Previous analysis of the two partial deletion alleles of *lin-42*, *n1089* and *ok2385* (Fig. 1A), have suggested the possibility that different developmental functions could be genetically separable. Although both deletions caused precocious heterochronic phenotypes, only *ok2385* showed evidence of slow and arrhythmic molting (Edelman et al, 2016; Monsalve et al, 2011; Tennessen et al, 2006). To confirm these findings, we quantified molt, intermolt, and larval stage duration for mutant and wild-type animals at high temporal resolution, using a luciferase assay that monitors entry and exit from the lethargus state associated with molting on animals grown in isolation (Meeuse et al, 2020; Olmedo et al, 2015). Indeed, we found that *lin-42(ok2385)* mutant larvae developed more slowly than wild-type animals and became increasingly arrhythmic (Figs. 1B and EV1A–C). Moreover, and as observed previously

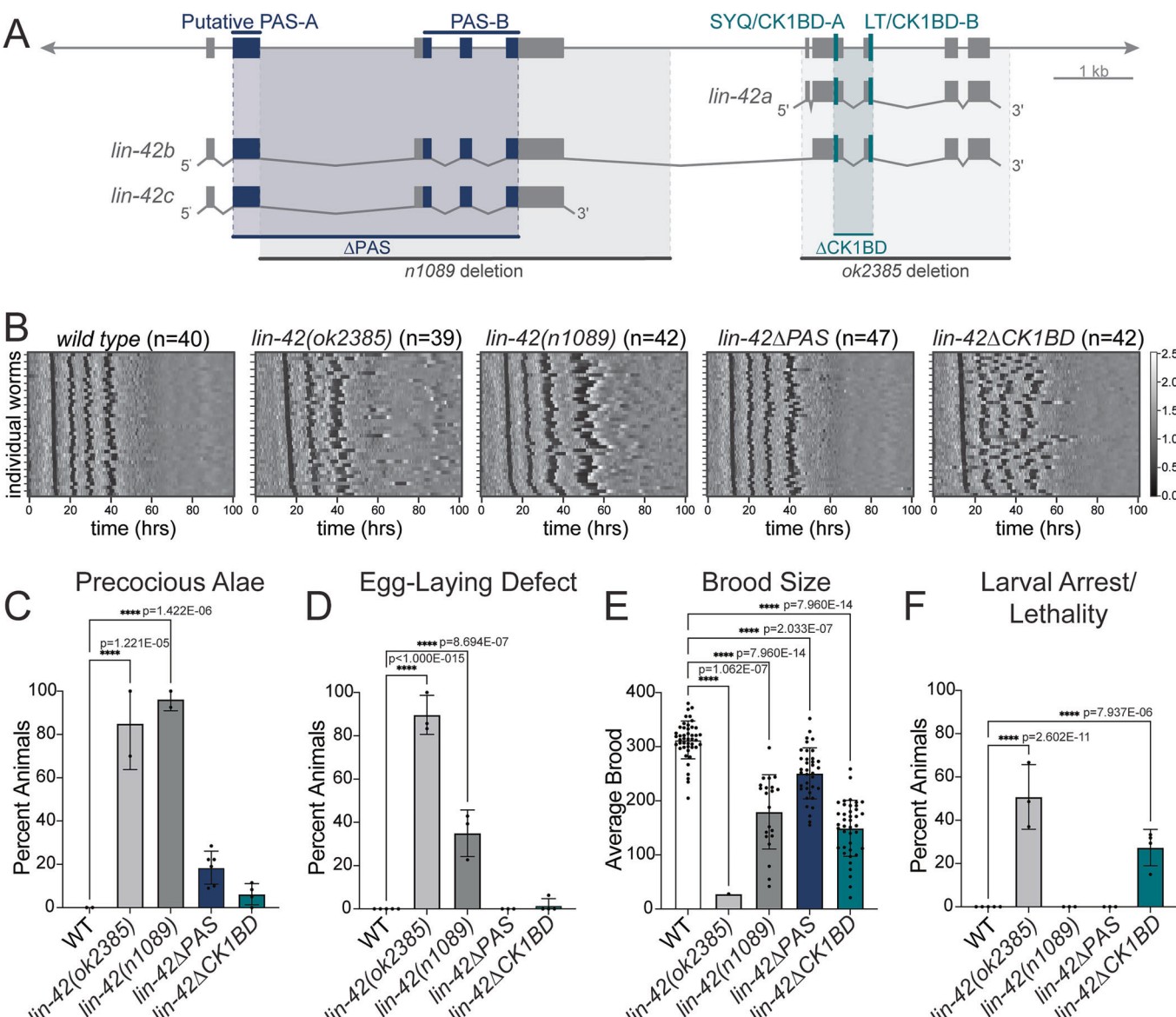

**Figure 1. *lin-42ΔPAS* mutations cause mild heterochronic phenotypes while *lin-42ΔCK1BD* mutations cause asynchronous molting.**

(A) Schematic depiction of the *lin-42* genomic locus and the major isoforms a, b and c (see also Appendix Fig. S2A). The location of sequences encoding putative PAS-A, PAS-B, SYQ, and LT sequence motifs are indicated. Published *lin-42* deletion alleles *n1089* and *ok2385* are indicated in dark and light grey, respectively, newly generated targeted *ΔPAS* and *ΔCK1BD* deletions in navy and teal, respectively. (B) Heatmaps showing trend-corrected luminescence traces for wild-type, *lin-42(ok2385)*, *lin-42(n1089)*, *lin-42(wrd67[ΔPAS])* and *lin-42(wrd63[ΔCK1BD])* animals as indicated. Each horizontal line represents one animal. Traces are sorted by entry into the first molt. Darker color indicates low luminescence signal and corresponds to the molts. Two biological replicates were performed for each strain. (C) Bar plot quantifying the percentage of animals of the indicated genotype with precocious complete or partial alae at the L3-L4 molt. n = 25 (*wild type*), 30 (*lin-42(ok2385)*), 36 (*lin-42(n1089)*), 52 (*lin-42(wrd67[ΔPAS])*), and 44 (*lin-42(wrd63[ΔCK1BD])*). Two biological replicates were performed, except for *lin-42(wrd67[ΔPAS])* for which three biological replicates were performed. (D) Bar plot quantifying the percentage of animals of the indicated genotype that exhibited egg-laying defects as determined by the presence of hatched larvae in the animal. n = 65 (*wild type*), 20 (*lin-42(ok2385)*), 64 (*lin-42(n1089)*), 69 (*lin-42(wrd67[ΔPAS])*), and 87 (*lin-42(wrd63[ΔCK1BD])*). Three biological replicates were performed, except for wild type and *lin-42(wrd63[ΔCK1BD])* for which four biological replicates were performed. (E) Bar plot depicting the average number of live progeny from hermaphrodites of the indicated genotype. n = 45 (*wild type*), 1 (*lin-42(ok2385)*), 21 (*lin-42(n1089)*), 36 (*lin-42(wrd67[ΔPAS])*), and 40 (*lin-42(wrd63[ΔCK1BD])*). Four biological replicates were performed, except for *lin-42(n1089)* and *lin-42(ok2385)*, for which three and one biological replicates were performed, respectively. The severe bagging phenotype of *lin-42(ok2385)* led to the exclusion of most animals from the broodsize analysis. (F) Bar plot quantifying the percentage of animals of the indicated genotype that arrested or died as larvae. n = 65 (*wild type*), 79 (*lin-42(ok2385)*), 73 (*lin-42(n1089)*), 69 (*lin-42(wrd67[ΔPAS])*), and (*lin-42(wrd63[ΔCK1BD])*). Three biological replicates were performed, except for wild type and *lin-42(wrd63[ΔCK1BD])* for which four biological replicates were performed. (C–F) Statistical significance was determined using an ordinary one-way ANOVA. P < 0.05 was considered statistically significant. **** indicates <0.0001. Error bars in (C–F) represent standard deviation and the center represents the mean. Source data are available online for this figure.

(Olmedo et al, 2015), most mutant animals underwent only three molts (33/39 animals) within the duration of this assay, suggesting either a precocious exit from the molting cycle, or a very delayed or abnormal fourth molt (Figs. 1B and EV1D). By contrast, essentially all *lin-42(n1089)* animals completed a normal number of molts (42/42 animals) and maintained robust synchrony, although they developed significantly slower than wild-type animals (Fig. 1B). At the same time, both alleles caused robust heterochronic phenotypes, illustrated by precocious formation of alae (Fig. 1C) in L3 to early L4 larvae. Alae are a cuticular structure normally secreted by the terminally differentiated epidermal seam cells at the L4-adult transition.

The complex nature of the alleles, each with at least one breakpoint in a noncoding region, precludes a straightforward attribution of the phenotypes to specific features of the LIN-42 protein. Therefore, we decided to generate precise deletions of the two conserved regions, PAS and SYQ-LT (Fig. 1A). Henceforth, and based on the functional characterization that we describe below, we will refer to SYQ-LT as CK1BD for CK1-Binding Domain. To our surprise, *lin-42(wrd67[ΔPAS])* mutant animals exhibited essentially wild-type molting patterns in the luciferase assay, resembling neither *lin-42(n1089)* nor *lin-42(ok2385)* mutant animals. By contrast, *lin-42(wrd63[ΔCK1BD])* mutant animals exhibited increasing loss of molting synchrony over time (Fig. 1B). However, unlike *lin-42(ok2385)*, *lin-42ΔCK1BD* mutant animals executed four detectable molts during the assay (Figs. 1B and EV1D).

To get a better understanding of the developmental relevance of each domain, we examined additional phenotypes (Fig. 1D–F; Appendix Table S1). We found that all four alleles affected brood sizes, albeit to varying degrees, in the order *ok2385»ΔCK1BD ≈ n1089 > ΔPAS* of decreasing severity (Fig. 1E; Appendix Table S1). Only *ok2385* and *n1089* caused highly penetrant egg-laying (Egl) and precocious alae defects (Edelman et al, 2016; Monsalve et al, 2011; Tennessen et al, 2006), whereas *ΔPAS* and *ΔCK1BD* mutants had completely wild-type egg laying and only a low penetrance of precocious alae (Fig. 1C,D; Appendix Table S1). Finally, both *ok2385* and, to a lesser extent, *ΔCK1BD* caused larval arrest and lethality, a phenotype interestingly not observed in the luciferase assay, suggesting an environmental dependency (Fig. 1F; Appendix Table S1). Taken together, these results indicate surprisingly mild phenotypes upon loss of the LIN-42 PAS region and highlight an important function of the CK1BD for rhythmic molting.

## LIN-42 and KIN-20 interact in vivo

Since the genetic results suggested separable functions of LIN-42, we hypothesized that these could depend on specific interaction partners. To identify interaction partners, we performed an anti-FLAG immunoprecipitation on endogenously tagged *lin-42(xe321[3xflag::lin-42)* mixed-stage animal lysates followed by mass spectrometry. As a control for non-specific binding, we also performed an anti-FLAG immunoprecipitation in an unrelated strain (*sart-3::gfp::3xflag*) (Rüegger et al, 2015). We observed a single highly enriched interaction partner for LIN-42: KIN-20—the orthologue of mammalian CK1δ/ε (Fig. 2A; Appendix Table S2). We did not detect an interaction between LIN-42 and NHR-85, previously observed in a yeast two-hybrid screen (Kinney et al, 2023).

The mRNA levels of both *lin-42* and *kin-20* oscillate (Hendriks et al, 2014; Meeuse et al, 2020). Further analysis of the published sequencing data revealed that the peak phases of the two RNAs differ, but that there are periods where both transcripts accumulate (Fig. 2B). Periods of co-expression were also observed for the proteins: we found by Western blotting that the levels of endogenously tagged KIN-20 protein were largely invariant over time, whereas those of endogenously tagged LIN-42 protein changed rhythmically (Fig. 2C). Both KIN-20 and LIN-42 occur in several isoforms that partially overlap in size (Fig. 2C; Appendix Fig. S2). We did not pursue the isoform identity of each band further. Finally, we confirmed a physical interaction with a reciprocal anti-HA pulldown from *lin-42(xe321[3xflag::lin-42]); kin-20(xe328[3xflag::ha::kin-20])* animal lysates, again using endogenously tagged proteins (Fig. 2D). Taken together, we conclude that LIN-42 and KIN-20 form a complex in vivo.

## The LIN-42 SYQ-LT regions form a functional CK1BD capable of binding mammalian CK1

Mammalian CK1 binds PER through its kinase domain (Crosby et al, 2023; Dahlberg et al, 2009). This region is highly conserved, with CK1δ and KIN-20 exhibiting 79% and 100% sequence identity within the kinase domain and active sites, respectively (Fig. EV2) (Spangler et al, 2025). By contrast, the CK1BD is less well conserved in LIN-42 with only ~30% sequence identity, different spacing of the relative LIN-42 CK1BD-A and -B sites, and no clear FASP-like sequence (Fig. 3A) (Spangler et al, 2025). Hence, it was unclear whether this region could function as a CK1BD in LIN-42.

We produced recombinant proteins to test this possibility. We were unable to express soluble KIN-20 from a bacterial or insect system, so we used recombinant human CK1δ kinase domain (hereafter referred to as CK1), similar to previous studies of KIN-20 (LaBella et al, 2020). In vitro pulldown assays with biotinylated LIN-42 protein containing the SYQ/CK1BD-A and LT/CK1BD-B domains as well as C-terminal tail (residues 402–589) revealed that CK1 bound to wild-type LIN-42 protein. Deletion of either the SYQ/CK1BD-A or LT/CK1BD-B motifs reduced this interaction (Fig. 3B).

To test whether the CK1BD is sufficient to bind CK1, we purified a truncated fragment of LIN-42 lacking the C-terminal tail (CK1BD) and performed bio-layer interferometry (BLI) assays with biotinylated CK1 and titrating LIN-42 fragments. This optical technique measures macromolecular interactions in real-time, allowing the extraction of kinetic parameters ($k_{on}$ and $k_{off}$) from model fits of the association and dissociation phases across a range of analyte concentrations. These parameters are then used to calculate the equilibrium dissociation constant ($K_D$, also known as an affinity constant). The amplitude of the BLI response is proportional to the mass of the bound complex and is therefore reflective of the molecular weight of the analyte. Accordingly, the CK1BD fragment produced a lower response signal compared to the CK1BD + Tail protein (residues 402–598; same construct used in Fig. 3B), reflecting the smaller size of the CK1BD alone rather than a difference in affinity. Global fitting across concentrations revealed that both constructs bound to biotinylated CK1 with comparable nanomolar affinities ($K_D = 303 \pm 107$ nM (CK1BD) and $309 \pm 153$ nM (CK1BD + Tail) (Fig. 3C–E).

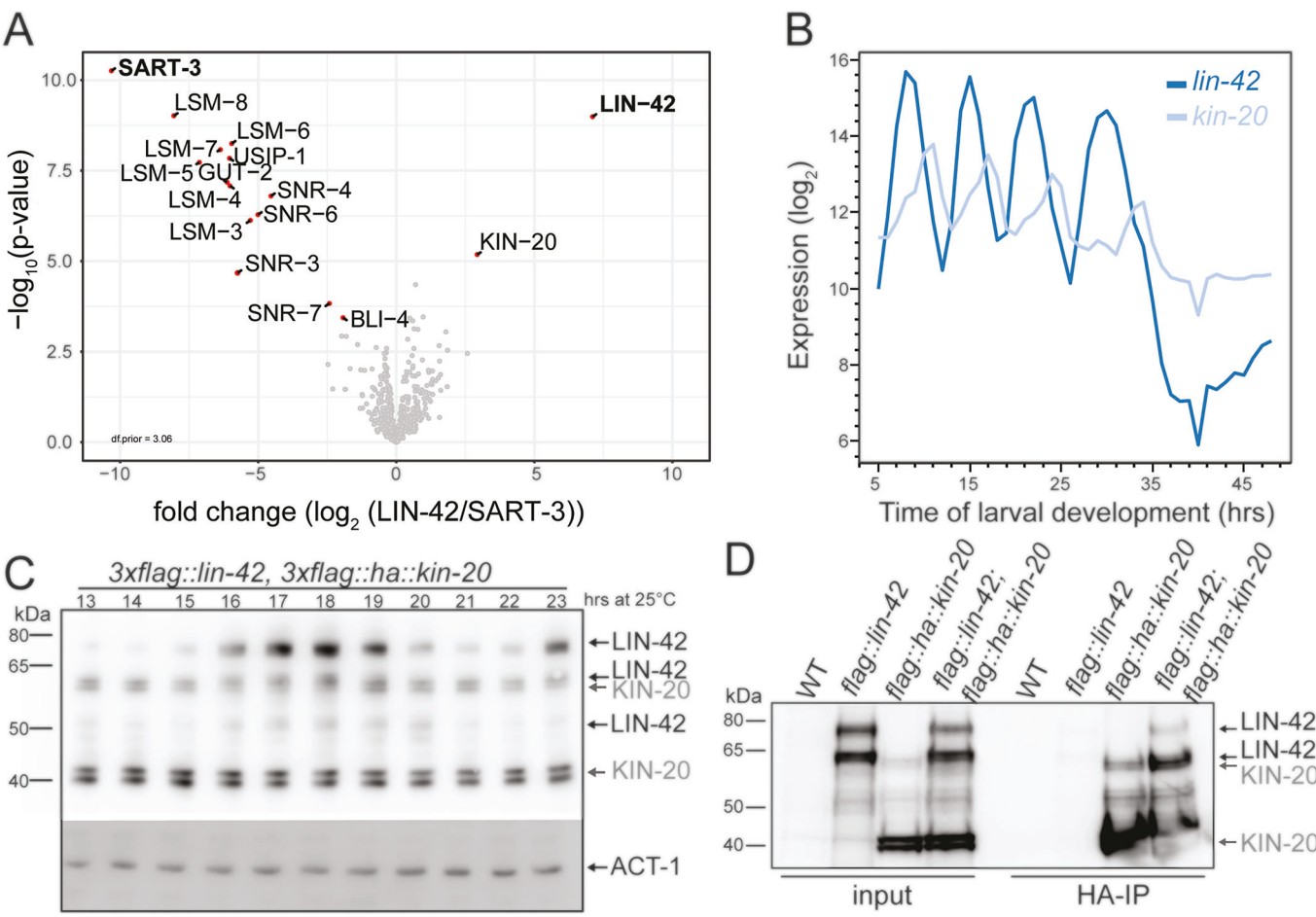

**Figure 2. LIN-42 and KIN-20 interact in vivo.**

(A) Volcano blot comparing protein enrichments in 3xFLAG::LIN-42 and 3xFLAG::SART-3 (control) immunoprecipitations, determined by mass spectrometry. The significantly enriched proteins are labeled with a red dot; see also Table S2. Two biological repeats were performed. (B) mRNA expression profile for *lin-42* (dark blue) and *kin-20* (light blue) mRNAs throughout larval development. Data Source: Meeuse et al, 2020. (C) Western blot with extracts from *3xflag::lin-42; 3xflag-ha::kin-20* (HW3479) larvae collected hourly for 11 h, starting at 13 h after plating synchronized L1 stage animals on food at 25 °C. Two biological repeats were performed. Top part probed with anti-FLAG-HRP (1:1000), lower part with anti-actin-1 (1:7500). LIN-42 and KIN-20 occur in several isoforms (indicated by arrows); see Appendix Fig. S2 for further information. (D) Anti-HA pulldown from synchronized animals (32 h at 25 °C) of the indicated genotype. Blot probed with anti-FLAG (1:1000). Three biological repeats were performed. Source data are available online for this figure.

We also purified an N-terminal LIN-42 construct that contains the single canonical PAS-B domain and the adjacent structured but unclassified element (N-terminus; residues 1–315) to confirm that this region, as in the mammalian system, is not involved in binding to CK1. We titrated LIN-42 N-terminus as high as 200 μM and, as expected, we did not observe any evidence of binding to biotinylated CK1 (Appendix Fig. S3). Taken together, these results demonstrate that the LIN-42 SYQ and LT motifs encompass a minimal CK1BD.

In mammals, the CK1BD region of PER1/2 is composed of two structured helical motifs (CK1BD-A and CK1BD-B) that facilitate stable binding to CK1. To confirm that the LIN-42 SYQ and LT regions function in a similar binding capacity as the CK1BD-A and CK1BD-B motifs, we performed BLI experiments with LIN-42 CK1BD + Tail protein in which the SYQ/CK1BD-A and LT/CK1BD-B motifs

were removed individually and in tandem. LIN-42-binding to CK1 was not detectable when CK1BD-B was deleted, either singly (*CK1BD-ΔB*) or in combination with CK1BD-A (*CK1BD-ΔA/B*) even at higher concentrations (Fig. 3C,F,G). In contrast, deletion of the CK1BD-A domain reduced the affinity of LIN-42 for CK1 approximately 10-fold ($K_D$ = ~3 μM) (Fig. 3C,H), necessitating higher concentrations of the CK1BD-ΔA + Tail protein to determine an accurate $K_D$ (Fig. 3H). These data indicate that CK1BD-A and B both contribute to high-affinity binding of CK1 in vitro, while revealing a more important contribution of the CK1BD-B motif, consistent with recent work that demonstrated an essential role of the human PER2 CK1BD-B motif for stable association with CK1 in a mouse model and mammalian cells (An et al, 2022). Collectively, our data establish that the LIN-42 CK1BD is functionally conserved and mediates stable binding to the human CK1 kinase domain.

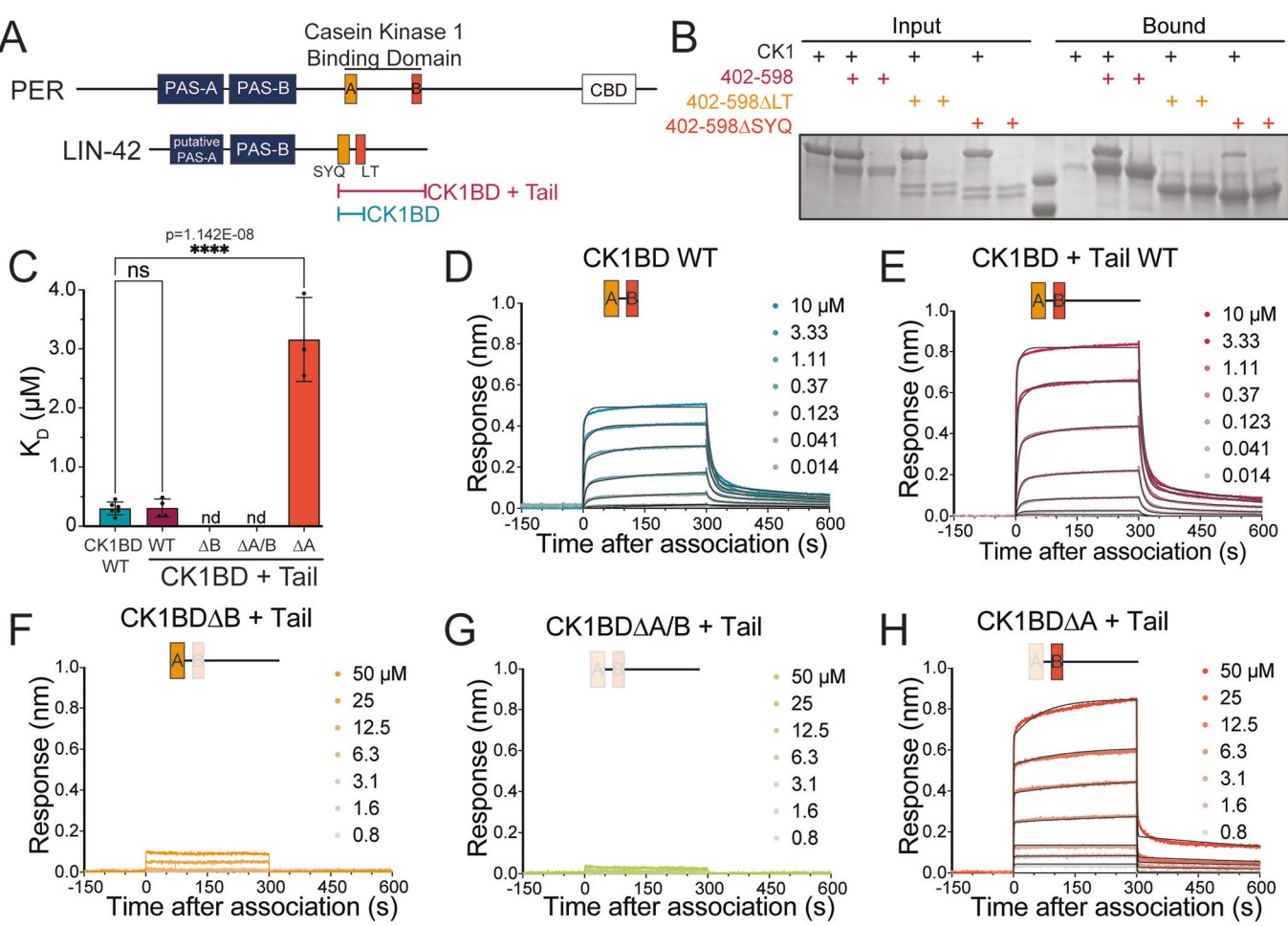

**Figure 3. The LIN-42 SYQ and LT regions constitute a functional CK1-binding domain.**

(A) Schematic representing PER2 and LIN-42 protein domains. Protein constructs used in this study, CK1BD + Tail and CK1BD, are indicated. CK1BD, Casein Kinase 1-Binding Domain; CBD, CRY-binding domain. (B) Representative pulldown assay of human CK1 and biotinylated LIN-42 CK1BD + Tail proteins using the indicated protein variants. Three technical replicates were performed. (C) Values for $K_D$ from kinetic analysis of BLI data (D–H) based on a 2:1 heterogeneous ligand binding model and global analysis (Octet). Mean ± SEM, ordinary one-way ANOVA. $P < 0.05$ was considered statistically significant. **** indicates $P < 0.0001$. (D–H) Bio-layer interferometry (BLI) data for indicated LIN-42 protein binding to immobilized, biotinylated CK1. Inset values represent the concentrations of LIN-42 for individual binding reactions. Model fit to association and dissociation over time is represented by black lines. Data shown from one representative experiment of $n \geq 3$ technical repeats. Source data are available online for this figure.

## Phosphorylation of the LIN-42 C-terminus by CK1 exhibits a conserved mode of feedback inhibition

Mammalian CK1 regulates PER abundance by controlling its degradation post-translationally. Mutations on CK1 or its PER phosphorylation sites can induce changes in circadian period as great as ~4-h (Appendix Fig. S1B) (Lowrey et al, 2000; Toh et al, 2001; Xu et al, 2005, 2007). To test whether LIN-42 is a CK1 substrate, we performed in vitro $^{32}$P-ATP kinase assays using the CK1BD alone or the CK1BD + Tail as substrates. CK1BD protein contains 4 serine and 6 threonine residues while the CK1BD + Tail construct contains an additional 22 serine and 15 threonine residues. We observed phosphorylation for both constructs, yet more robustly on LIN-42 CK1BD + Tail than on CK1BD alone (Fig. 4A–C). Hence, CK1 appears capable of phosphorylating both CK1BD and the extended C-terminal tail. Deletion of CK1BD-B reduced phosphorylation relative to the wild-

type protein, whereas phosphorylation of the CK1BD-ΔA + Tail mutant protein was largely comparable to that of the wild-type protein (Fig. 4A–C). Thus, in the two assays, loss of CK1BD-B impacts both binding and phosphorylation by human CK1 more strongly than loss of CK1BD-A (Figs. 3C–H and 4A–C).

To explore the kinetics of CK1-mediated phosphorylation of the LIN-42 extended C-terminus, we performed substrate titration experiments using an ADP-Glo enzymatic assay. Here, we observed a decrease in kinase activity with high levels of wild-type LIN-42 CK1BD + Tail (Fig. 4D). This result mirrors the ability of phosphorylated PER as well as the apoptosis substrate p63, to inhibit the activity of CK1 (DBT in Drosophila) in both mammalian and *Drosophila* systems through product inhibition (Gebel et al, 2020).

Closer examination revealed that deletion of the LIN-42 CK1BD-A motif relieved feedback inhibition (Fig. 4D), explaining why the $^{32}$P-ATP data had shown little decrease in the overall

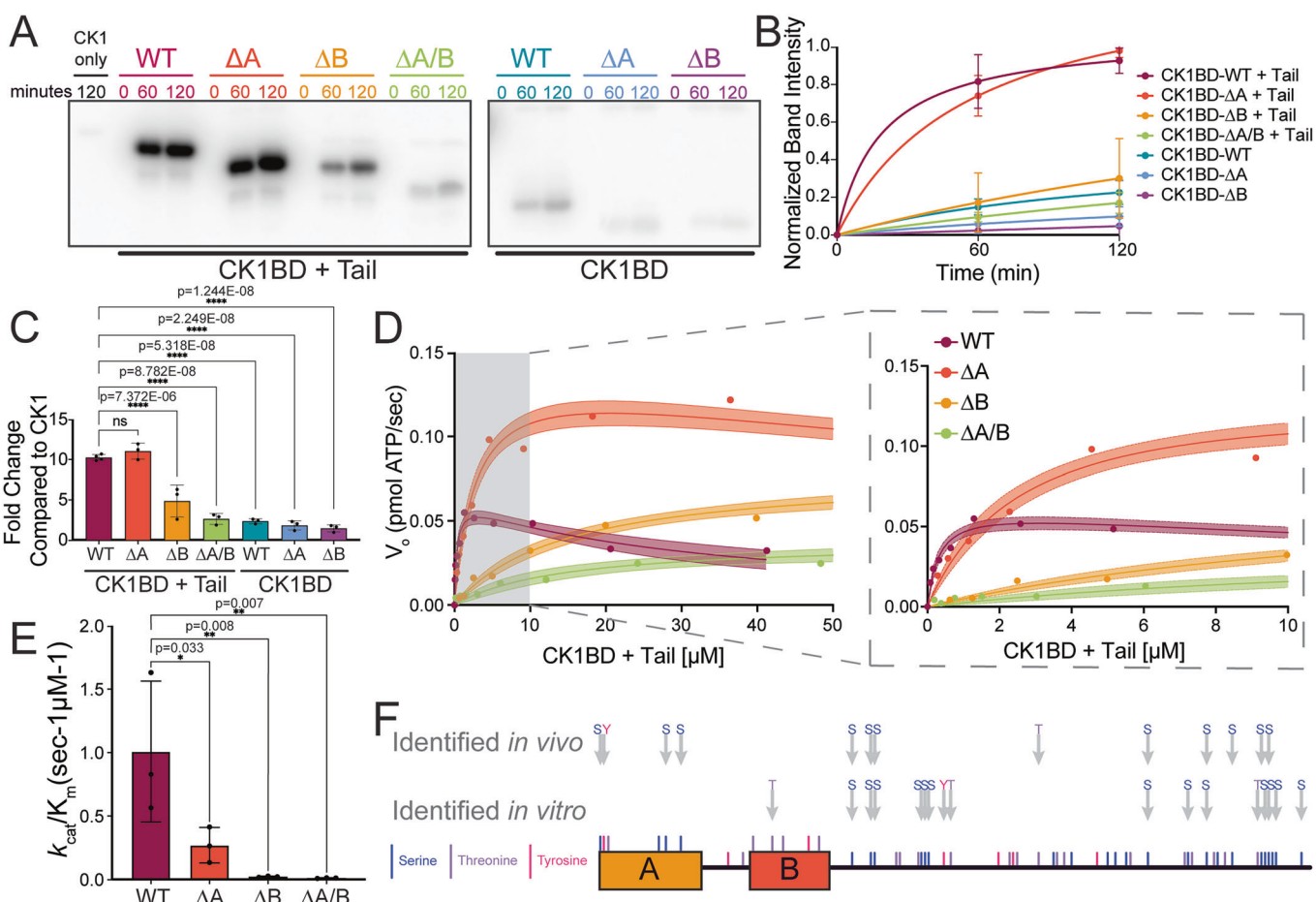

**Figure 4. The extended C-terminus of LIN-42 is phosphorylated by CK1.**

(A) Representative ${}^{32}$P-radiolabeled ATP enzymatic assay of CK1 activity on LIN-42 CK1BD + Tail and CK1BD wild-type and indicated domain deletion mutants. Three technical replicates were performed. (B) Quantification of band intensity depicted in (A) and replicates. Model fit to Michaelis–Menten, $n = 3$. (C) Fold-change quantification of band intensity from (A, B) compared to CK1 autophosphorylation at 120 min. Ordinary one-way ANOVA. $P < 0.05$ was considered statistically significant. ****$P < 0.0001$, $n \geq 3$, mean ± SEM. (D) Representative ADP-Glo enzymatic assay comparing activity of CK1 on wild-type LIN-42 CK1BD + Tail wild-type (maroon), CK1BD-ΔA (orange), CK1BD-ΔB (yellow), and CK1BD-ΔA/B (green). Three technical replicates were performed. (E) Values of kcat/Km calculated from (D). $n = 3$, mean ± SEM, Ordinary one-way ANOVA. $P < 0.05$ was considered statistically significant. * indicates $P < 0.03$ and ** indicates $P < 0.002$. (F) LIN-42 phosphorylation sites identified via mass spectrometry from in vivo samples from whole worm lysate collected hourly from synchronized L1 animals from 14 h (early L2) to 25 h (late L3) after plating at 25 °C (top arrows) and in vitro kinase reactions of CK1 activity on LIN-42 CK1BD + Tail (bottom arrows). Three technical replicates were performed for the in vitro assay and the in vivo assay was performed once. Serine, threonine, and tyrosine residues are indicated via vertical blue, purple, and pink ticks, respectively, along the CK1BD + Tail construct schematic. Source data are available online for this figure.

phosphorylation level (Fig. 4A–C) despite reduced affinity for the kinase (Fig. 3C). These data suggest to us that the SYQ/CK1BD-A motif contributes to an inhibitory mechanism, possibly by stabilizing a non-productive complex between CK1 and phosphorylated substrate, thereby limiting product turnover. Without SYQ/CK1BD-A, product release may be facilitated, enabling more sustained CK1 activity despite weaker binding. Such a mechanism would be consistent with recent studies identifying a conserved mode of CK1 product inhibition (Gebel et al, 2020; Philpott et al, 2023; Harold et al, 2024). Conversely, deletion of the B motif either in isolation or with the A motif, caused a significant reduction in overall phosphorylation of CK1BD + Tail (Fig. 4D). No serine and only 4 threonine residues are removed in this deletion, so this effect was likely not due to loss of phosphosites. The catalytic efficiency ($k_{cat}/K_m$) of CK1 for each LIN-42 substrate decreased significantly

with deletion of the CK1BD motifs relative to wild-type, with activity most severely impacted upon loss of the CK1BD-B motif (Fig. 4D,E). Taken together, these results suggest that while both motifs are involved in binding to CK1, deletion of the CK1BD-A region predominantly relieves feedback inhibition without fully disrupting the anchoring interaction seen with the deletion of the CK1BD-B domain. Moreover, and similar to the mammalian PER–CK1 complex (Marzoll et al, 2022), loss of the anchoring interaction compromises, but does not fully abrogate phosphorylation.

## Identification of potential LIN-42 phosphosites

To determine potential CK1-dependent LIN-42 phosphorylation sites, we performed in vitro kinase reactions followed by

phosphoenrichment and mass spectrometry to identify phospho-peptides. There are a total of 26 serine and 21 threonine residues in the CK1BD + Tail construct. Of these, 12 serine and 3 threonine sites in the C-terminus were phosphorylated upon incubation of LIN-42 CK1BD + Tail with CK1 at least once in three replicates in vitro (Fig. 4F). Although we detected low levels of phosphorylation on LIN-42 constructs lacking the tail in the $^{32}$P-ATP assay (Fig. 4A), no serine and only 1 threonine phosphorylation within the CK1BD was detected in vitro, reflecting possible limitations in the assays.

To test whether LIN-42 was phosphorylated in vivo, we immunoprecipitated endogenously tagged 3xFLAG::LIN-42 at various time points from a population of synchronized L2 stage larvae and subjected precipitates to mass spectrometry. We identified 15 serine residues, 1 threonine residue, and 1 tyrosine residue that were phosphorylated on LIN-42. All but four of these phosphosites (all serine residues) are located on the CK1BD + Tail (Fig. 4F). Among these, 6 serine residues overlap with the 13 phosphoserines identified after the in vitro reaction while no threonine or tyrosine residues were phosphorylated in both the in vitro and in vivo datasets (Fig. 4F). The overlap of in vitro and in vivo results is consistent with the possibility that LIN-42 is a CK1 substrate in vivo. Whether incomplete overlap reflects the activity of additional kinases, phosphatases, and/or specificity factors in vivo, or technical differences in the experimental approaches remains to be determined.

## LIN-42 C-terminal tail deletion causes a heterochronic phenotype but leaves molting timing largely unaffected

Given the extensive phosphorylation of the LIN-42 tail in vitro and in vivo, we sought to explore its functional relevance. Unexpectedly, *lin-42ΔTail* mutant animals resembled *lin-42(n1089)* animals phenotypically and did not recapitulate the dramatic arrhythmic molting phenotype seen with *lin-42ΔCK1BD* (Figs. 1C–F and 5). Specifically, *lin-42ΔTail* mutants exhibited regular timing of the first three molts, yet an extended and irregular fourth molt (Figs. 5A and EV3E). The severity of this extended 4th molt was variable and could reflect unaccounted environmental differences during assay runs. The mutant animals also exhibited a modest reduction in brood size, egg-laying defects and precocious alae but no larval arrest (Fig. 5B–E). *lin-42(n1089)* is a partial deletion with a breakpoint in a noncoding region of *lin-42* (Fig. 1A)(Tennessen et al, 2006) and it is currently unknown what mature transcripts and proteins are made in these mutant animals. Thus, it is unclear why *lin-42ΔTail* animals phenotypically resemble *lin-42(n1089)*. These data indicate that the LIN-42 C-terminus is less relevant for timing of larval molting and suggest instead that it plays a previously unrecognized role in the heterochronic functions of LIN-42.

## *kin-20* null and catalytically dead mutants exhibit arrhythmic molting

Deletion of a major phosphorylation target, the LIN-42 tail, did not recapitulate *lin-42ΔCK1BD* mutant molting timing defects. This prompted the question whether KIN-20 was at all required for rhythmic molting. Hence, we monitored molt timing for *kin-20(ok505)* null mutant animals (*kin-20(0)*). *kin-20* loss recapitulated both the slow development and arrhythmic molt phenotypes of *lin-*

42(ok2835) and *lin-42ΔCK1BD* mutations, but with more pronounced arrhythmicity (Figs. 1B, 5A, and EV3A–C). Like *lin-42(ok2835)* and *lin-42ΔCK1BD* mutations, the *kin-20(0)* mutation also caused larval arrest when animals were grown on the plate but not in the liquid culture luciferase assay (Figs. 1F and 5E). Additionally, we found that *kin-20(0)* mutant animals resembled *lin-42(ok2835)* mutants in their severe egg-laying defects and brood size reduction, phenotypes that are weaker or even absent from *lin-42ΔCK1BD* animals (Figs. 1D,E and 5B,C). Finally, like *lin-42ΔCK1BD* animals but unlike *lin-42(ok2385)* mutants, *kin-20* null mutants did not develop precocious alae (Figs. 1C and 5D), consistent with earlier work (Banerjee et al, 2005).

To assess the relevance of KIN-20's enzymatic activity for rhythmic molting, we engineered a D310A point mutation into the endogenous *kin-20* locus to abrogate catalytic activity. This mutation disrupts the HRD motif in the catalytic loop that coordinates phosphorylated residues (Fig. EV2) (Johnson et al, 1996; Nolen et al, 2004). The resulting *kin-20(xe355[D310A])* animals recapitulated the *kin-20(0)* mutant arrhythmic molting phenotype, although KIN-20 levels were not decreased (Figs. 5A and EV3A–C; Appendix Fig. S4A–C). Indeed, we observed a trend towards increased accumulation of the mutant KIN-20 protein, possibly indicating autoregulation but not further pursued by us (Appendix Fig. S4A). We conclude that KIN-20 and especially its enzymatic activity are required for rhythmic molting.

## KIN-20 exhibits dynamic changes in subcellular localization

Although KIN-20 kinase activity and its binding to the LIN-42 CK1BD are required for rhythmic molting, LIN-42 tail phosphorylation appeared largely dispensable. Moreover, *kin-20* deletion or inactivation appeared to cause more pronounced defects than deletion of the LIN-42 CK1BD. Hence, we wondered whether the significance of the interaction between the two proteins could lie in regulation of KIN-20 by LIN-42, rather than vice versa. We therefore examined KIN-20 levels and localization. Using GFP::KIN-20 (with a split-GFP system to enhance the signal; Methods), we observed both nuclear and cytoplasmic signals in the epidermis. Strikingly, the relative distribution between these compartments varied with developmental stage. Microfluidics-based observation of early larvae (Berger et al, 2021) revealed substantial cytoplasmic GFP::KIN-20 signal during molts and at the beginning of larval stages but a predominantly nuclear signal in the middle of the larval stage (Fig. EV4). We recapitulated these dynamics in L4 stage animals grown on plates (Fig. 6A), which additionally revealed an apparent membrane-bound pool of KIN-20 in seam cells (Fig. 6A).

## The LIN-42 CK1BD and C-terminal tail promote nuclear accumulation of KIN-20

The timing of increased nuclear KIN-20 in the middle of a larval stage coincides with peak accumulation of LIN-42 (Fig. 2C). Moreover, using endogenously tagged GFP::LIN-42 and wrmScarlet::KIN-20, we found that the two proteins co-localized in the nuclei of epidermal seam and hyp7 cells (Fig. 6B). Hence, we wondered if LIN-42 contributed to the change in KIN-20 accumulation. To test this notion, we examined how the lack of LIN-42-binding affected KIN-20 accumulation dynamics.

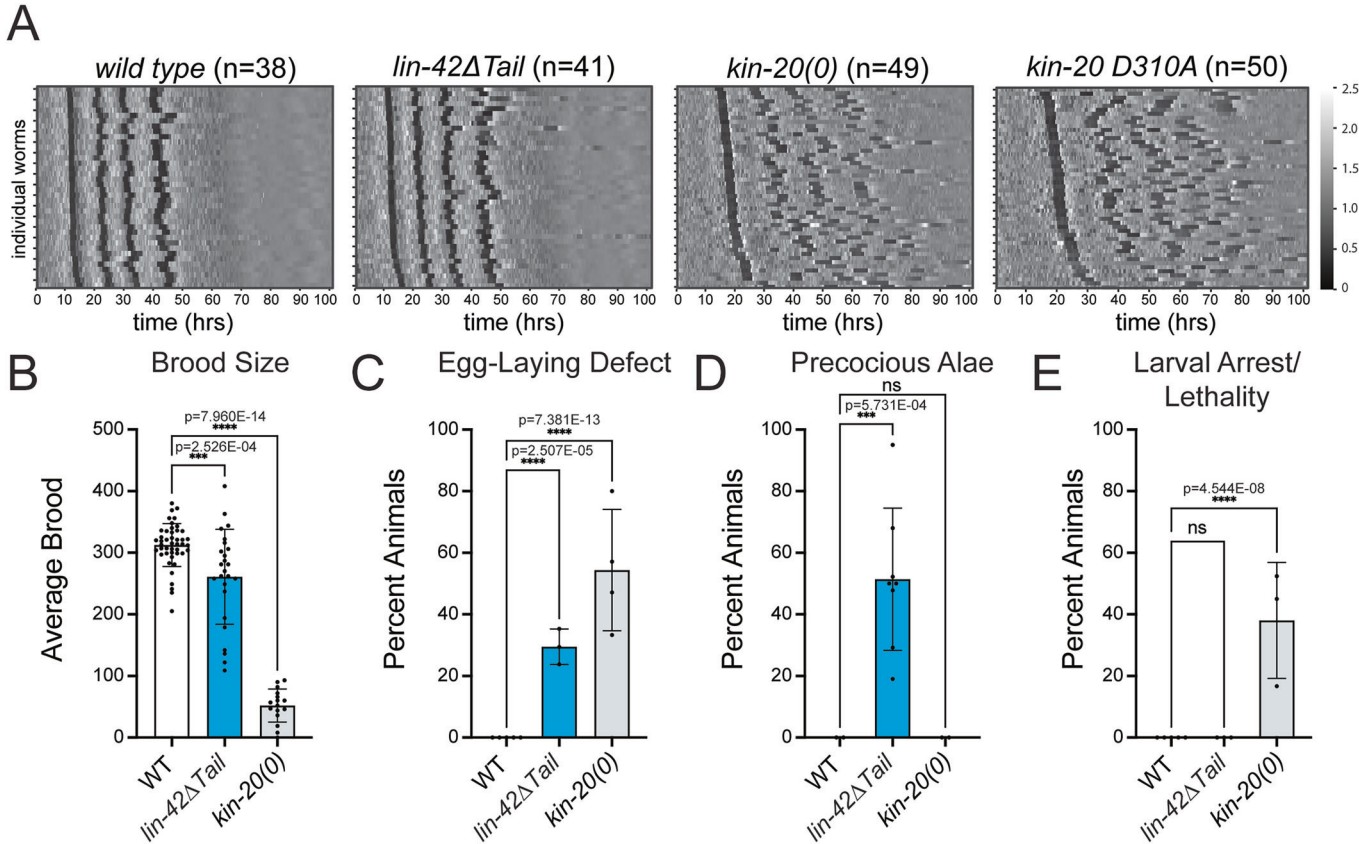

**Figure 5.** *kin-20* null and catalytically dead alleles exhibit larval arrest and asynchronous molting similar to *lin-42ΔCK1BD* mutants.

(A) Heatmaps showing trend-corrected luminescence traces from wild-type, *lin-42(wrd107[ΔTail])*, *kin-20(0)*, and *kin-20(xe355[D310A])* animals. Each horizontal line represents one animal. Traces are sorted by entry into the first molt. Darker color indicates low luminescence signal and corresponds to the molts. Two biological replicates were performed for each strain. (B) Bar plot depicting the average number of live progeny from hermaphrodites of the indicated genotype. n = 45 (*wild type*), 25 (*lin-42(wrd107[ΔTail])*), 16 (*kin-20(0)*). The following number of biological replicates were performed: wild type = 4; *lin-42(wrd107[ΔTail])* = 3; *kin-20(0)* = 2. (C) Bar plot quantifying the percentage of animals of the indicated genotype that exhibited egg-laying defects as determined by the presence of hatched larvae in the animal. n = 65 (*wild type*), 55 (*lin-42(wrd107[ΔTail])*), 46 (*kin-20(0)*). Four biological replicates were performed, except for *lin-42(wrd107[ΔTail])*, for which three biological replicates were performed. (D) Bar plot quantifying the percentage of animals of the indicated genotype with precocious complete or partial alae at the L3-L4 molt. n = 25 (*wild type*), 112 (*lin-42(wrd107[ΔTail])*), 38 (*kin-20(0)*). Two biological replicates were performed, except for *lin-42(wrd107[ΔTail])*, for which five biological replicates were performed. (E) Bar plot quantifying the percentage of animals of the indicated genotype that arrested or died as larvae. n = 65 (*wild type*), 63 (*lin-42(wrd107[ΔTail])*), 80 (*kin-20(0)*). Three biological replicates were performed, except for wild type, for which four biological replicates were performed. (B–E) Statistical significance was determined using ordinary one-way ANOVA. P < 0.05 was considered statistically significant. ** indicates P < 0.0001, **** indicates P < 0.0001. Error bars in (B–E) represent standard deviation. Source data are available online for this figure.

Specifically, we visualized GFP::KIN-20 *in lin-42ΔCK1BD* mutant animals during mid-L4 stage. We observed reduced nuclear GFP signal relative to wild-type animals (Fig. 6C,E) in both hypodermal and seam cell nuclei and little change in cytoplasmic levels. We also visualized GFP::KIN-20 in *lin-42ΔTail* mutant animals. Surprisingly, this deletion also reduced KIN-20 nuclear accumulation similar to the *lin-42ΔCK1BD* mutants (Fig. 6D,E). These data indicate that LIN-42 promotes nuclear accumulation of KIN-20, but that additional mechanisms may contribute to the extensive arrhythmia of *lin-42ΔCK1BD* molting.

## Discussion

PER and its nematode orthologue LIN-42 control two distinct types of biological timing: mammalian and fly circadian rhythms (PER)

and *C. elegans* developmental progression (LIN-42). The identification of additional orthologous pairs functioning in these respective pathways, ROR/NHR-23, CK1/DBT/KIN-20, Timeless/TIM-1, and REV-ERB/NHR-85, suggests that there is a conserved set of biological timing genes, whose function can be exploited in different contexts. While these factors are conserved across circadian and developmental timing systems, their interactions can vary dramatically as can the timers that they control. For instance, NHR-23 exhibits conserved timekeeping functions as it is essential for larval molting (Kostrouchova et al, 1998; Johnson et al, 2023; Kostrouchova et al, 2001; Patel et al, 2022) and was also recently shown to be a critical driver of ~24-h rhythmic gene expression in adult *C. elegans* (Hiroki and Yoshitane, 2024). In circadian rhythms the NHR-23 ortholog RORA has an antagonistic relationship with the REV-ERB nuclear hormone receptor whereas in *C. elegans* developmental timing NHR-23 and the REV-ERB

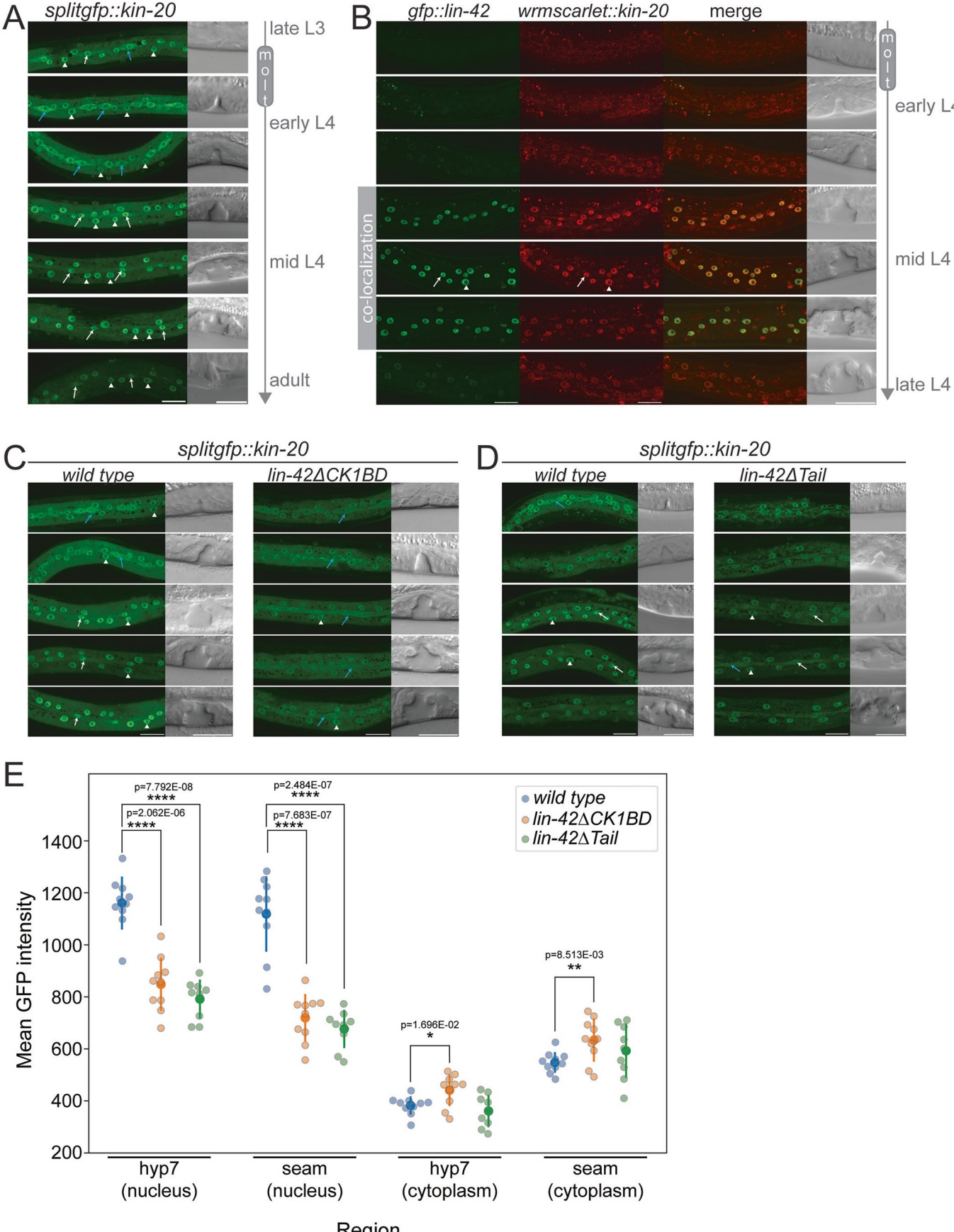

**Figure 6.    The LIN-42 CK1BD and C-terminal tail are required for KIN-20 nuclear accumulation.**

(A) Confocal images of *splitgfp::kin-20* ((*bchSi84[eft-3p::gfp1-10(codon-optimized)::tbb-2 3UTR]); kin-20(xe354[3xflag::4xgfp11])*) animals staged according to vulva morphology (Mok et al, 2015) from late L3 to adult. Arrows indicate seam cell nuclear (white) and cytoplasmic (blue) localization; white arrowheads indicate nuclear hyp7 localization. Scale bar = 20 μm. Two biological repeats were performed. (B) Confocal images of *lin-42(xe315[gfp::tev::3xflag::lin-42]) kin-20(xe329[wrmScarlet::tev::lin-ker::kin-20])* staged according to vulva morphology. An arrow indicates seam cell, an arrowhead hyp7 nuclear localization. Scale bar = 20 μm. Two biological repeats were performed. (C) Confocal images of *splitgfp::kin-20* in wild-type and *lin-42(wrd63[ΔCK1BD])* animals during early–mid L4 stage, based on vulva morphology. Arrows indicate seam cell nuclear (white) and cytoplasmic (blue) localization, white arrowheads indicate hyp7 nuclear localization. Scale bar = 20 μm. Three biological repeats were performed. (D) Confocal images of *splitgfp::kin-20* in *wild-type* and *lin-42(wrd107[ΔTail])* animals during early–late L4 stage based on the vulvae morphology. Arrows indicate seam cell nuclear (white) and cytoplasmic (blue) localization, white arrowheads indicate hyp7 nuclear localization. Scale bar = 20 μm. Two biological repeats were performed. (E) Quantification of GFP::KIN-20 signal in hyp7 (nucleus), seam cells (nucleus), hyp7 (cytoplasm) and seam cells (cytoplasm) of wild-type (blue, $n = 10$), *lin-42(wrd63[ΔCK1BD])* (orange, $n = 10$) and *lin-42(wrd107[ΔTail])*(green, $n = 9$) animals. Each condition with mid-L4 stage worms. Statistical significance was determined using a $t$ test. $P < 0.05$ was considered statistically significant. ** indicates $P < 0.0001$****, indicates $P < 0.0001$. Mean ± SD. Source data are available online for this figure.

ortholog act cooperatively to promote *lin-4* microRNA expression. Our work here shows the first example of a conserved functional module in these clocks: the LIN-42/PER – KIN-20/CK1 complex.

## The PAS domains are largely dispensable for LIN-42 developmental timing function

The PAS domains are the most notable feature of LIN-42 sequence conservation and were thought to be an interaction platform for a developmental stage-specific binding partner (Jeon et al, 1999). Previous work characterizing the *lin-42(ok2385)* and *n1089* alleles suggested that the PAS domains were necessary for the hetero-chronic functions of LIN-42 (Monsalve et al, 2011; Tennessen et al, 2006). Yet, our precise deletion of the LIN-42 PAS domains caused only a mild precocious alae phenotype with significantly less penetrance than in *lin-42(n1089)* mutants, suggesting that this model is incorrect. While *lin-42(n1089)* removes the sequence of the PAS domains, it also results in a truncation of *lin-42b* and removal of part of the *lin-42a* promoter sequence (Monsalve et al, 2011; Tennessen et al, 2006). The *n1089* phenotype could reflect loss of the PAS domains combined with potential reduction in expression of other elements such as the C-terminal tail or an NHR-85 interaction site (Kinney et al, 2023).

The weak phenotypes seen with our precise deletion are particularly striking when considering that the PAS domains, and specifically the highly conserved PAS-B domain, can mediate LIN-42 dimerization (Lamberti et al, 2024)—a function that is important for PER activity in circadian rhythms (Zheng et al, 1999). Although it remains to be determined whether the PAS domains are at all necessary for LIN-42 dimerization in vivo, it seems possible that LIN-42's developmental timing function does not require dimerization.

## LIN-42 contains a CK1BD that is important for rhythmic molting

Contrasting with the PAS domain, we found the LIN-42 SYQ/LT regions to be required for rhythmic molting, and we could demonstrate in vitro that they constitute a functional CK1BD that anchors CK1, promotes LIN-42 phosphorylation, and mediates kinase inhibition. Moreover, the patterns of phosphorylation that CK1 generates on LIN-42 in vitro overlap with those that we observed on LIN-42 in vivo. We note that it remains to be demonstrated to which extent the in vivo phosphorylation events depend on KIN-20. Our attempts to address this question by

examining LIN-42 phosphorylation in strains lacking active KIN-20 were thwarted by the slow growth, sickness, and arrhythmic molting observed in these animals.

The two CK1BD subdomains have separable functions in vitro. CK1BD-B plays a more prominent role in kinase binding, mirroring the anchoring function of its mammalian and *Drosophila* equivalents (An et al, 2022; Marzoll et al, 2022; Philpott et al, 2023). Anchoring allows CK1 to target lower affinity, non-consensus motifs that are key regulatory sites in fly/mammalian circadian clocks (An et al, 2022; Kim et al, 2007; Marzoll et al, 2022; Nawathean et al, 2007; Philpott et al, 2023). However, without the PER CK1BD-B, not all capacity for phosphorylation is lost, suggesting that high-affinity binding is not essential for phosphorylation but promotes it (Marzoll et al, 2022). For LIN-42 CK1BD-ΔB, we also see low levels of phosphorylation by CK1 in vitro (Fig. 4), suggesting that it may play a similar role in controlling KIN-20 activity.

The LIN-42 CK1BD-A plays a less prominent role in binding but rather is involved in CK1 inhibition. CK1δ-dependent phosphorylation of both mammalian and *Drosophila* PER at sites near the CK1BD leads to feedback inhibition of CK1 through conserved anion binding sites (Philpott et al, 2023). Our in vitro LIN-42 phosphorylation data reveal a decrease in enzyme catalytic activity as substrate concentration increases, dependent on the CK1BD-A, consistent with feedback product inhibition of CK1 (Fig. 4). Deletion of the LIN-42 CK1BD-A motif relieved the inhibition, presumably by supporting release of phosphorylated product and thereby preventing trapping of the enzyme in an inactive complex. Further enzymatic and structural studies will be required to distinguish between these modes of regulation. Although our experiments used human CK1 for technical reasons, given the high degree of conservation between CK1 and KIN-20 including complete conservation of the active site (Fig. EV2), we predict that these results will be transferable to *C. elegans* KIN-20.

## Function of the LIN-42-CK1 interaction

Many models of circadian clock function highlight phosphorylation of PER by CK1 as the central consequence of their interaction, controlling PER stability and thereby setting the period length. Intriguingly, although CK1 heavily phosphorylates the LIN-42 C-terminal tail, including on serine and threonine residues that are conserved among *Caenorhabditis* nematodes, but not on more distantly related nematodes (Fig. EV5), deletion of this sequence impacts molting rhythmicity only modestly, affecting mostly the

fourth molt. These data suggest either phosphosite redundancy, or that regulated phosphorylation of LIN-42 is not the key function of this complex for controlling molting timing. Consistent with the latter notion, loss of KIN-20 or its catalytic activity impair rhythmic molting much more dramatically than loss of binding to LIN-42 through the CK1BD.

For PER–CK1, several lines of evidence show that reciprocal to CK1-mediated regulation of PER, PER also regulates CK1 activity, both on itself and potentially other targets such as CLOCK. Indeed, PER and CRY function as a bridging complex to provide CK1 access to CLOCK (Aryal et al, 2017; Cao et al, 2021), and mutation of the CK1BD causes defects in PER and CK1 nuclear accumulation as well as CK1-mediated phosphorylation of CLOCK (Cao et al, 2021; Chiou et al, 2016). Product inhibition through phosphory-lated PER may thus limit phosphorylation not only of PER itself, but potentially also of other substrates, including CLOCK. Strikingly, KIN-20 also exhibits dynamic subcellular accumulation, with nuclear accumulation depending on LIN-42 binding. Whether this nuclear accumulation reflects co-transport, nuclear retention, or stabilization of the nuclear KIN-20 pool is not known. We note that reducing nuclear KIN-20 levels appears insufficient to explain the arrhythmic molting phenotype of *lin-42ΔCK1BD* animals since such a reduction also occurs in *lin-42ΔTail* animals that exhibit rhythmic molting. Hence, we speculate that it is the combination of promoting nuclear KIN-20 accumulation and regulating its activity that explains the importance of the CK1BD for rhythmic molting. Such a scenario may also explain the previous observation that forced expression of the short LIN-42a isoform, largely comprising the CK1BD + Tail, caused extended molt durations (Monsalve et al, 2011), as this might sequester KIN-20 and/or inhibit its kinase activity. However, it remains possible that dynamic and regulated nuclear accumulation of KIN-20 is not at the heart of the LIN-42–KIN-20 complex function.

### Distinct functions of LIN-42 and KIN-20 in molting versus heterochronic timing?

Our data not only support the existence of LIN-42 and KIN-20 in a stable complex that is important for molting timing but also suggest that LIN-42 and KIN-20 can function independently of one another in some contexts. Neither *kin-20(lf)* nor *lin-42ΔCK1BD* mutations cause precocious alae, a phenotype observed with several other *lin-42* mutant alleles, including *n1089* and deletion of the LIN-42 C-terminal tail.

These findings agree with previous genetic data that suggested independent functions of *lin-42* and *kin-20* in regulating the expression of the heterochronic *let-7* microRNA (Rhodehouse et al, 2018). A recent study on LIN-42's function in regulating another heterochronic miRNA, *lin-4*, indicated that physical interaction with the NHR-85 transcription factor may support LIN-42's activity in this process, and perhaps the heterochronic pathway more generally (Kinney et al, 2023). Relatively mild effects upon *nhr-85* deletion, and LIN-42's ability to interact with additional transcription factors in a yeast two-hybrid assay suggest that additional functionally relevant interactions partners remain to be identified. Exploring the relative contributions of the structured N-terminus and C-terminal tail to the heterochronic functions of LIN-42 are a particularly interesting future direction.

The apparent differences in wiring among components of the heterochronic pathway and the molting timer may reflect their different modes of timekeeping where the heterochronic pathway is chiefly concerned with the order of events and the molting timer—similar to the circadian clock—with the tempo and/or robustness of their execution. Indeed, although LIN-42/PER have previously been suggested to provide an evolutionary link between circadian (PER) and heterochronic (LIN-42) timing systems (Jeon et al, 1999), our data suggest that the link primarily derives from functional conservation between circadian and molting timing. Hence, we may speculate that the heterochronic function of LIN-42, which appears to involve neither rhythmic activity nor conserved interactions with orthologues of other circadian clock proteins (Banerjee et al, 2005; Gissendanner et al, 2004; Hasegawa et al, 2005; Jeon et al, 1999; Kostrouchova et al, 1998; Migliori et al, 2023) arose secondarily to, or independent from, a rhythmic timing function. Finally, we note with interest that the phosphorylation of PER by CK1 is an important component of temperature compensation in the circadian clock (Narasimamurthy and Virshup, 2021) that allows this clock to maintain its period despite variation in temperature. By contrast, the period of *C. elegans* molting and oscillatory gene expression changes with temperature. Our data that the LIN-42 tail, where the bulk of CK1 mediated phosphorylation occurs, is dispensable for rhythmic molting may thus reflect the lack of robust temperature compensation in the *C. elegans* developmental clock.

## Methods

**Reagents and tools table**

| Reagent/resource | Reference or source | Identifier or catalog number |
|---|---|---|
| **Experimental models** | | |
| *lin-42(wrd63) II* | This work | JDW335, short: *lin-42(ΔCK1BD)* |
| *lin-42(wrd107) II* | This work | JDW439, short: *lin-42(ΔTail)* |
| *lin-42(wrd67) II* | This work | JDW340, short: *lin-42(ΔPAS)* |
| *lin-42(ok2385)* | CGC | RB1843 |
| *lin-42(n1089)* | CGC | MT2257 |
| *kin-20(ok505)* | CGC | VC398, short: *kin-20(0)* |
| *xeSi312 [eft-3p::luc::gfp::unc-54 3' UTR, unc-119(+) in oxTi177] IV* | This work | HW1993, short: *luciferase* reporter strain |
| *lin-42(wrd67) II; xeSi312 [eft-3p::luc::gfp::unc-54 3' UTR, unc-119(+) in oxTi177] IV* | This work | JDW409, short: *lin-42ΔPAS* (in *luciferase* background) |
| *lin-42(wrd63) II; xeSi312 [eft-3p::luc::gfp::unc-54 3' UTR, unc-119(+) in oxTi177] IV* | This work | JDW417, short: *lin-42ΔCK1BD* (in *luciferase* background) |
| *lin-42(wrd107) II; xeSi312 [eft-3p::luc::gfp::unc-54 3' UTR, unc-119(+) in oxTi177] IV* | This work | JDW590, short: *lin-42ΔTail* (in *luciferase* background) |

| Reagent/resource | Reference or source | Identifier or catalog number |
|---|---|---|
| lin-42(ok2385) II xeSi312 [eft-3p::luc::gfp::unc-54 3' UTR, unc-119(+) in oxTi177] IV | This work | JDW658, short lin-42(ok2385) (in luciferase background) |
| lin-42(n1089) II; xeSi312 [eft-3p::luc::gfp::unc-54 3' UTR, unc-119(+) in oxTi177] IV | This work | HW3730, short lin-42(n1089) (in luciferase background) |
| xeSi312 [eft-3p::luc::gfp::unc-54 3' UTR, unc-119(+) in oxTi177] IV; kin-20 (ok505) X | This work | HW3368, short: kin-20(0) (in luciferase background) |
| xeSi312 [eft-3p::luc::gfp::unc-54 3' UTR, unc-119(+) in oxTi177] IV, kin-20(xe355[kin-20 D310A]) X | This work | HW3484, short: kin-20 D310A (in luciferase background) |
| lin-42 (xe321[3xflag::lin-42])II, kin-20 (xe328[3xflag::ha::kin-20]) X | This work | HW3479, short: flag::lin-42; flag::ha::kin-20 |
| lin-42(xe315[gfp::tev::3xflag::lin-42]) II; kin-20(xe329[wrmScarlet::tev::linker::kin-20]) X | This work | HW3303, short: gfp::lin-42; wrmScarlet::kin-20 |
| EG6701, xeSi55[dpy-30p::sart-3::gfp::his::flag::xrn-2 3', unc-119 (+)] I | Rüegger et al (2015) | HW1008, short: sart-3::gfp::flag |
| lin-42 (xe321[3xflag::lin-42]) II | This work | HW3270, short: flag::lin-42 |
| xeSi312 [eft-3p::luc::gfp::unc-54 3' UTR, unc-119(+)in oxTi177] IV; kin-20(xe401[3xflag::ha::kin-20 D310A]) X | This work | HW3859, short: kin-20(xe401[3xflag D310A]) (in luciferase background) |
| xeSi312 [eft-3p::luc::gfp::unc-54 3' UTR, unc-119(+)in oxTi177] IV; kin-20(xe400[3xflag-ha::kin-20 D310A]) X | This work | HW3858, short: kin-20(xe400[3xflag D310A]) (in luciferase background) |
| (bchSi84[eft-3p::gfp1-10(codon optimized)::tbb-2 3UTR]) II; kin-20(xe354[3xflag::4xgfp11]) X | This work | HW3451, short: splitgfp::kin-20 |
| lin-42 (wrd63)II; bchSi84 (pIK407[Peft-3::gfp1-10(codon-optimized)::tbb-2 3UTR]) II; Kin-20(xe354[3xflag::4xgfp11::kin-20]) X | This work | JDW792, short: lin-42(ΔCK1BD); splitgfp::kin-20 |
| kin-20(xe328[3xflag::ha::kin-20]) | This work | HW3293, short: flag::ha::kin-20 |
| bchSi84 (pIK407[Peft-3::gfp1-10(codon-optimized)::tbb-2 3UTR]) II | This work | IFM217 |
| lin-42(wrd107) II; bchSi84 (pIK407[Peft-3::gfp1-10(CO)])II; kin-20(xe354[3xflag::4xgfp11]) X | This work | HW3892, short: lin-42(ΔTail), splitgfp::kin-20 |

| Oligos used | Reference or source | Identifier or catalog number |
|---|---|---|
| **Repair templates for CRISPR-Cas 9 injections** | | |
| ACTGACCCGAGAAGCACTGA CACTGCACACTAAACGGTTCGAGGA TGAATATAAGGACACTTGGTG CAGACTCCGAGATTCTCAGAATT AATAAGCTACTGCCCA | This work | 6920 |

| Reagent/resource | Reference or source | Identifier or catalog number |
|---|---|---|
| GAAAGTTGCCAGCGCCCCGC CGACCACCTC ACTTGGTGCAGGTG AGAGAATTTTC TGAGTTATTT | This work | 6158 |
| CACTTGCACCAGCAATGCG TGAGGAAGGTGCCACGCT CAAGGATCAGAACCAGGGC TTCCCGGCCAACAT | This work | 5783 |
| GAATAATATATATTCAAA TTTTCAGCGGAGATGGACT ATAAAGACGATGATGA CAAAGATTACAAGGACGA CGACGACAAAGACTAC AAAGATGATGACGACAAG GGAGGTGGAGGTGGAGCT TACCCATACGATGTTCC AGATTACGCTGGAGGTG GAGGTGGAGCTGAACTTC GTGTCGGCAATCG TTTCCGCCTCGG | This work | KK107 |
| TCCCGCTATTTTCCTATTA AAATCTTTCTTCAACTCTT ATTTTATTCCAGAACGTGG CACCATCAGCCAAATGG ACTACAAAGACCATGAC GGTGATTATAAAGATCA TGACATCGATTACAAGG ATGACGATGACAAGGAGC CAGCCGGGCACTCAAG CGCAACACATAACATCG TTGTGCCCAACGCC AATCCCACGC | This work | KK90 |
| TGAAAACCGTCCTGCTG CTTGCCGATCAAATGTTGTCT CGTGTGGAATTTATTCATT GCCGAGATTACATTCATCG CGCGATTAAGCCGGATAAC TTTTTAATGGGTCTTGGAA AACGAGGAAATCTGGTCTA TGTAAGTTTTTCTTTT GTGAGGGATTAGCCAGCCT ACTATGTTGTGCC TTTTTTTGCAGATTATTGA | This work | KK137 |
| **Primers for amplification of repair template from a plasmid** | | |
| TGCTTTTTAAAACCAAATTTCCC GCTATTTTCCTATTAAAATCT TTCTTCAACTCTTATTTT ATTCCAGAACGTGGCAC CATCAGCCAAATGAGT AAAGGAGAAGAACTT TTCACTGGAGT | This work | KK41 |
| CGCGAGGAGCTAGGCAGGCT AGAAGGCAAACTGTACCTG CGTGGGATTGGCGTTGGG CACAACGATGTTATGTGTTGC GCTTGAGTGCCCGGCTGGC TCCATGCTTCCGCCG GTACCTCC | This work | KK42 |
| CTTCTTCATTTGTTTGAAT ATTTTGACCCAAGTAGATGT CACCGAACGAGCCGCTTCC GATTTTGCGACCGAGGCG GAAACGATTGCCGACAC GAAGTTCGCTTCCGCC GGTACCTCCAC | This work | KK102 |

| Reagent/resource | Reference or source | Identifier or catalog number |
|---|---|---|
| GCTTTACGTCAATGTCAG AGCGATTTGAAATCTAAAG TGCAAAATTCACAATAC CAGATTTATGAATTGTGAAT AATATATATTCAAATTTTC AGCGGAGATGGTCA GCAAGGGAGAGGC | This work | KK103 |
| GTGAATAATATATATTCAAAT TTTCAGCGGAGATGAGT ACCTCCGGCGGATCCGG | This work | KK145 |
| CGACCGAGGCGGAAACGAT TGCCGACACGAAGTTCGC TTCCGCCGGTACCTCCAC | This work | KK146 |
| **Genotyping primers** | | |
| CATCTTGCCATCATCACCAC | This work | 4529 |
| TGGGTTCCGATAGAATTTGG | This work | 4530 |
| CCAGTCCCTTTTGCCTGGAT | This work | 5790 |
| TGGGTTCCGATAGAATTTGGCAT | This work | 5791 |
| GGGAGGCAGTGTGTCAAAAC | This work | 6921 |
| GGCTTGAATGTTTGGGCCTG | This work | 6922 |
| GTCCTGAATTGGCCTGAAAA | This work | 5784 |
| ATTCTCTCACCTGCACCAAG | This work | 5785 |
| ACCGTGGGGTAATGTGAAGG | This work | 5786 |
| CTGTGTGAAGTTTTGGCATCT | This work | KK43 |
| CTTCCTCACGCATTGCTG | This work | KK44 |
| AGATACCTCCAGTTCCGC | This work | KK104 |
| TCGAGTGAAGGGCCTAGT | This work | KK105 |
| TTCGTGTCGGCAATCGTTT | This work | KK117 |
| ATTCGCTCATTTCGGCTC | This work | KK139 |
| **Plasmids** | | |
| linker::gfp::tev::3xflag | This work | pIK384 |
| wrmScarlet::tev::linker::aid::3xflag | This work | pIK385 |
| 4xgfp11::linker::tev::3xflag | This work | pIK401 |
| eft-3p::gfp1-10 (codon optimized)::tbb-2 3'UTR | This work | pIK407 |
| eft-3p::luc::gfp::unc-54 3' UTR | Meeuse et al (2020) | pMM002 |
| **crRNAs** | | |
| GAGTGGTGGGTCCGTTGAGG | This work | 143 |
| CCTCCTCTCTCCTAATGCTA | This work | 144 |
| AAGACGAGTACAAGGACACT | This work | 145 |
| GGTGTTCGGGGGTGACAACG | This work | 146 |
| CCACCATCACTCAAGCCTCA | This work | 147 |
| GTACTCGTCTTCAAATCGCT | This work | 359 |
| TTGATGGGTCTTGGAAAGCG | This work | KK136 |
| GCGGAGATGGAACTTCGTGT | This work | KK99 |
| TCAGCCAAATGGAGCCAGCC | This work | KK50 |

| Reagent/resource | Reference or source | Identifier or catalog number |
|---|---|---|
| **Recombinant DNA plasmids** | | |
| pET-22b(+) HNXL-LIN-42 CK1BD (402–475) | This work | CP1149 |
| pET-22b(+) HNXL-LIN-42 CK1BD + Tail (402–598) | This work | CP1213 |
| pET-22b(+) HNXL-LIN-42 CK1BDΔA + Tail (434–598) | This work | CP1433 |
| pET-22b(+) HNXL-LIN-42 CK1BDΔB + Tail (402-445,468-598) | This work | CP1434 |
| pET-22b(+) HNXL-LIN-42 CK1BDΔA/B + Tail (434-445, 468-598) | This work | CP1435 |
| pET-22b(+) HisGβ1-LIN-42 N-terminus (1–315) | This work | CP491 |
| pET-22b(+) HisGST-CK1δΔC (1–317) | Philpott et al (2023) | CP1020 |
| His-TEV | Blommel and Fox (2007) | N/A |
| **Antibodies** | | |
| monoclonal ANTI-FLAG® M2-peroxydase (HRP) antibody produced in mouse | Sigma-Alderich | A8592 |
| Mouse Anti-Actin Antibody, clone C4 | Millipore | MAB1501 |
| Mouse IgG HRP Linked Whole Ab | GE Healthcare | NXA931-1ML |
| **Oligonucleotides and other sequence-based reagents** | | |
| crRNA | This study | Table S4 |
| Genotyping Primers | This study | Table S3 |
| Repair templates for CRISPR genome editing | This study | Table S3 |
| Plasmids | This study | Table S3 |
| **Chemicals, enzymes and other reagents** | | |
| cOmplete™, EDTA-free Protease Inhibitor Cocktail | Roche | 11873580001 |
| Anti-FLAG® M2 Magnetic Beads | Millipore | M8823-1ML |
| Pierce™ Anti-HA Magnetic Beads | Thermo Scientific | 88837 |
| ECL™ Prime Western Blotting Detection Reagent | Cytiva | RPN2236 |
| Alt-R S.p. Cas9 Nuclease V3, 100 ug | Integrated DNA Technologies (IDT) | 1081058 |
| Alt-R CRISPR-Cas9 tracrRNA, 5 nmol | Integrated DNA Technologies (IDT) | 1072532 |
| Levamisole | Fluka Analytical | 31742 |
| NuPAGE™ Bis-Tris Mini Protein Gels, 4–12%, 1.0–1.5 mm | Invitrogen | NP0321BOX and NP0329BOX |
| Immun-Blot PVDF Membrane, Roll, 26 cm×3.3 m | Bio-Rad | 1620177 |

| Reagent/resource | Reference or source | Identifier or catalog number |
|---|---|---|
| PageRuler™ Prestained Protein Ladder, 10 to 180 kDa | Thermo Scientific | 26616 |
| Escherichia coli DH5α | NEB | C2987H |
| Escherichia coli BL21 (DE3) Rosetta2 | Fisher | 69041 |
| ADP-Glo | Promega | V9102 |
| Protease Inhibitor Cocktail | Fisher Scientific | 501657350 |
| Pierce Universal Nuclease | Fisher Scientific | PI88701 |
| Ni-NTA resin | Qiagen | 30230 |
| Glutathione Sepharose 4B resin | GE | 17-0756-05 |
| Precision Plus Protein Dual Color Standards | Bio-Rad | 1610374 |
| Isopropyl-β-D-thiogalactopyranoside | Fisher | BP-1755-100 |
| Streptavidin Magnetic Beads | Thermo Fisher | 88816 |
| Streptavidin Biosensors | Sartorius Corporation | 18-5020 |
| Trypsin/Lys-C Mix, Mass Spec Grade | Promega | V5073 |
| D-Luciferine free acid | PJK GmbH | 102112 |
| 384-well plate, white | Berthold | 32505 |
| **Software** | | |
| Napari | https://napari.org/stable/ | 0.5.3 |
| ImageJ | https://imagej.net/ij/ | 1.54 |
| PRISM | GraphPad | 10.4.2 (534) |
| Excel | Microsoft | 16.97 |
| Data Analysis HT | Octet | 11.0 |
| Illustrator | Adobe | 29.5.1 |
| **Other** | | |
| FastPrep-24™ 5 G bead beating grinder and lysis system | MP Biomedicals | 6005500 |
| Amersham Imager 680 blot and gel imager | Amersham | Imager 680 |
| Bioruptor® Plus sonication | Diagenode | B01020001 |
| Luminometer Centro XS³ LB960 | Berthold | 46970-50 |

## Methods and protocols

### C. elegans culture

*C. elegans* were grown and maintained at 20 °C unless indicated otherwise and cultured as described (Brenner, 1974) on MYOB (Church et al, 1995) or NGM 2% agar plates with *Escherichia coli* OP50 bacteria. Full genotypes of strains used in this work and the respective growth conditions are listed in Appendix Table S2. All mutant strains were backcrossed to N2 at least 2×. The Bristol N2 isolate was used as the wild type. Below, we describe how we generated novel strains for this study. In Appendix Table S3, we list oligo repair templates, plasmids and genotyping primers. In Appendix Table S4, we list crRNAs used.

### Genome editing

Novel *lin-42* mutant alleles (*wrd67[ΔPAS]*), (*wrd63[ΔCK1BD]*) and (*wrd107[ΔTail]*) using CRISPR-Cas9 were generated by injection of Cas9 ribonucleoprotein complexes as described (Ragle et al, 2022). Specified deletions were made in the endogenous *lin-42* locus in N2 animals. Repair templates were used at 100 ng/μl along with Cas9 at 250 ng/μl, crRNA oligos at 60 ng/μl, and 10 ng/μl co-injection marker (pCFJ90)(Frøkjaer-Jensen et al, 2008). Strains used in the luciferase assays were generated by crossing the mutant strains with a luciferase reporter strain on chromosome IV (HW1993, generated as previously described in Meeuse et al, 2020, using plasmid pMM002) (Appendix Table S3).

Strains containing endogenously tagged *lin-42* and *kin-20* were obtained using CRISPR/Cas9 as described previously (Ghanta and Mello, 2020). In short: 5 μg of Alt-R S.p. Cas9 Nuclease V3 (IDT, Cat # 1081058), 2 μg of Alt-R® CRISPR-Cas9 tracrRNA (IDT, Cat # 1072532) and 1.5 μg of crRNA were incubated at 37 °C for 15 min to form the RNP complex. 500 ng repair template and co-injection markers pIK127 (10 ng/μl) and pRF4 (*rol-6(su1006)*) (40 ng/μl) were added to a final volume of 20 μl in water. Mix was injected into N2 animals.

For *lin-42*, a *gfp::3xflag* (amplified from plasmid pIK384 with primers KK41/KK42) or *3xflag* tag (Ultramer DNA Oligo, KK90) was inserted at the N-terminus of LIN-42 isoforms.

*For kin-20*, a *3x::flag-ha* (Ultramer DNA oligo, KK107), *4xgfp11::linker:.flag* (amplified from plasmid pIK401 with primers KK145/KK146) or *wrmScarlet* (amplified from plasmid pIK385 with primers KK102/KK103) was inserted at the first common shared exon of all KIN-20 isoforms with following flanking sequence: 5′-tatattcaaattttcagCGGAGATG-[insert]-GAACTTCGTGTCGGCAATCGTTTCC-3′).

To reconstitute GFP with the split-GFP system, we crossed the obtained *4xgfp11::linker::flag::kin-20* line into a MosSCI strain containing the *gfp1-10* fragment expressed from an *eft-3* promoter. For this strain the *gfp1-10* sequence from plasmid pCZGY2254 (Noma et al, 2017) was codon optimized for *C. elegans*. After adding two synthetic introns the codon-optimized sequence was ordered as a gBlock from IDT. The sequence was cloned into a MosSCI-compatible backbone together with *eft-3*p promoter and *tbb-2* 3′UTR sequences to create plasmid pIK407 by Gibson assembly. pIK407 (*eft-3p::gfp1-10(codon-optimized)::tbb-2* 3′UTR) was inserted into the ttTi5605 site on chromosome II by MosSCI (Frøkjær-Jensen et al, 2012).

To generate a catalytic dead KIN-20, the D310A mutation was introduced by CRISPR-Cas9 into N2 and *3xflag-ha::kin-20* animals, respectively. Injection was performed as described previously for the *lin-42* and *kin-20* tagged strains. The ultramer DNA oligo KK137 containing the D310A mutation and two silent mutations (to abrogate crRNA recognition) was used as a repair template.

### Immunoprecipitation

For the 3xFLAG-LIN-42 IP, mixed stage animals were obtained by plating eggs after a standard hypochlorite treatment and growth at different temperatures (15 °C, 20 °C and 25 °C). 24 h after plating, worms were collected by pooling 30,000 worms from each

temperature condition per strain and replicate (triplicates for each strain). For the HA-KIN-20 IP, synchronized worms were collected after 32 h at 25 °C. For the mapping of LIN-42 phosphorylation sites, synchronized worms were grown at 25 °C and collected hourly from 14 h (L2 larval stage) to 25 h (L4 larval stage). Worm extracts and immunoprecipitation were done as previously described (Gudipati et al, 2021). In brief, extracts were made in lysis buffer (50 mM Tris-HCl, pH 7.5, 150 mM NaCl, 1% TRITON X-100, 1 mM EDTA) supplemented with 1 mM PMSF and 1 tablet of cOmplete Protease inhibitor (Roche, cat. No. 11 873 580 001) per 50 ml using an MP Biomedical Fast-Prep-24 5 G bead beater. For the FLAG IP, cleared worm extract was incubated with 50 μl of 50% bead suspension of anti-FLAG M2 Magnetic Beads (Sigma, Catalog Number M8823) for 2 h at 4 °C. For the HA-IP worm extract was incubated with 50 μl of bead suspension of Pierce™ Anti-HA Magnetic Beads (Thermo Scientific, Cat. No. 88837).

### Proteolytic digest

The beads were incubated for 4 h at RT with 800 rpm shaking in 5 μl of digestion buffer (3 M guanidine hydrochloride, 20 mM EPPS (4-(2-Hydroxyethyl)-1-piperazinepropanesulfonic acid) pH 8.5, 10 mM CAA (2-Chloroacetamide), 5 mM TCEP (Tris(2-carboxyethyl)phosphine hydrochloride)) and 1 μl of Lys-C (0.2 μg/μl in 50 mM HEPES, pH 8.5). 1 μl of trypsin (0.2 μg/μl) was added and incubated at 37 °C overnight. After 12 h another 1 μl of trypsin was added and incubation continued for 4 more hours at 37 °C.

### Mass spectrometry to identify LIN-42 interacting factors

The generated peptides were acidified with 0.8% TFA (final concentration) and analyzed by LC–MS/MS on an EASY-nLC 1000 [Thermo Scientific] with a two-column set-up. The peptides were applied onto a peptide 75 μm × 2 cm PepMap Trap trapping column [Thermo Scientific] in 0.1% formic acid, 2% acetonitrile in H2O at a constant pressure of 800 bar. Using a flow rate of 150 nl/min, peptides were separated on a 50 μm × 15 cm PepMap C18, 2 μm, 100 A [Thermo Scientific] at 45 °C with a linear gradient of 2%–6% buffer B in buffer A in 3 min followed by a linear increase from 6 to 22% in 40 min, 22–28% in 9 min, 28–36% in 8 min, 36–80% in 1 min, and the column was finally washed for 14 min at 80% buffer B in buffer A (buffer A: 0.1% formic acid; buffer B: 0.1% formic acid in acetonitrile). The separation column was mounted on an EASY-Spray™ source [Thermo Scientific] connected to an Orbitrap Fusion LUMOS [Thermo Scientific]. The data were acquired using 120,000 resolution for the peptide measurements in the Orbitrap and a top T (3 s) method with HCD fragmentation for each precursor and fragment measurement in the ion trap according to the recommendation of the manufacturer (Thermo Scientific). For the analysis protein identification and relative quantification of the proteins was performed with MaxQuant v.1.5.3.8 using Andromeda as search engine, and label-free quantification (LFQ). The C. elegans subset of the UniProt v.2021_05 combined with the contaminant database from MaxQuant was searched and the protein and peptide FDR were set to 0.01. The LFQ intensities estimated by MaxQuant were analyzed with the einprot R package (https://github.com/fmicompbio/einprot) v0.5.4. Features classified by MaxQuant as potential contaminants or reverse (decoy) hits or identified only by site, as well as features identified based on a single peptide or with a score below 10, were filtered out. The LFQ intensities were log2 transformed and missing values were imputed using the 'MinProb' method from the imputeLCMD R

package v2.0 with default settings. Pairwise comparisons were performed using limma v3.50.0, considering only features with at least 2 non-imputed values across all the samples in the comparison. Estimated log2-fold changes and P values (moderated t test) from limma were used to construct volcano plots.

### Mass spectrometry to map in vivo LIN-42 phosphorylation sites

The peptides from each sample were labeled with TMTpro reagents and pooled. The TMT labeled peptide mixture was subjected to off-line high pH fractionation on a YMC Triart C18 0.5 × 250 mm column (YMC Europe GmbH) using the Agilent 1100 system (Agilent Technologies). A total of 96 fractions was collected for each experiment and concatenated into 48 fractions. For each LC-MS analysis, all available peptides were loaded onto a PepMap Neo trap (Thermo Fisher) using the Vanquish Neo UHPLC system (Thermo Fisher). On-line peptide separation was performed on a 15-cm EASY-Spray™ C18 column (ES75150PN, Thermo Fisher) by applying a linear gradient of increasing ACN concentration at a flow rate of 200 nL/min. Orbitrap Fusion Lumos Tribrid (Thermo Fisher) mass spectrometer was operated in the data-dependent mode. The ions for the survey scan were collected for a maximum of 50 ms to reach the standard AGC target value and the scan recorded using an Orbitrap detector at a resolution of 120,000. The topmost intense precursor ions from the Orbitrap survey scan recorded every 3 s were selected for stepped higher-energy C-trap dissociation (HCD) at 29%, 32% and 35% normalized collision energy scan. To reach an AGC value of 100,000 ions, the maximum ion accumulation time for the MS2 scan was set to 500 ms. The TMT reporter ions were quantified using an MS2 scan recorded using the Orbitrap analyzer at a resolution of 50,000. Thermo RAW files were processed using Proteome Discoverer 2.4 software (Thermo Fisher) as described in the manufacturer's instructions. Briefly, the Sequest search engine was used to search the MS2 spectra against the C. elegans UniProt database (downloaded on 05/2023) supplemented with common contaminating proteins. For peptide identification, cysteine carbamidomethylation and TMTpro tags on lysine and peptide N-termini were set as static modifications, whereas oxidation of methionine residues and acetylation protein N-termini were set as variable modifications. The assignments of the MS2 scans were filtered to allow 1% FDR. For reporter quantification, the S/N values were corrected for isotopic impurities of the TMTpro reagent using the values provided by the manufacturer. The sums across all TMTpro reporter channels were normalized assuming equal total protein content in each sample.

### Western blot

Synchronized animals were collected hourly from 13 to 23 h after plating at 25 °C. Extracts were made by boiling at 95 °C for 5 min in lysis buffer (63 mM Tris-HCl (pH 6.8), 5 mM DTT, 2% SDS, 5% sucrose) followed by sonication with a BioRupter Plus (Diagenode) with the following settings: 13 cycles, 30 s on/off at 4 °C. Samples were cleared by centrifugation, before separating proteins by SDS-PAGE (loading: 50 μg protein extract per well) and transferring them to PVDF membranes by semi-dry blotting. The following antibodies were used: Monoclonal mouse anti-FLAG M2-Peroxidase (HRP) (Sigma-Aldrich; A8592, dilution: 1:1000), monoclonal mouse anti-Actin clone C4 (Millipore; MAB1501, dilution 1:7500) and anti-mouse HRP conjugated antibody (GE Healthcare

#NXA931, dilution 1:2000). The membrane was incubated with ECL™ Prime Western-Blot-Reagent (Cytiva, #RPN2236) and bands detected with an Amersham Imager 680 (GE Healthcare).

### Phenotypic analysis and microscopy

For phenotypic analyses, gravid adults were bleached, and embryos were hatched at low density onto MYOB plates seeded with OP50, or gravid adults were picked onto seeded plates and removed after 1–2 h. Animals were maintained at 20 °C and observed daily or twice daily. Bag-of-worms phenotype was determined when live progeny were observed inside adult animals. For brood counts, animals were individually transferred to fresh wells daily for 5 days after L4 and live progeny were counted and averaged.

To score premature alae, synchronized animals were collected from MYOB plates by washing off plates. In all, 1000 μl of M9 + 2% gelatin was added to the plate or well, agitated to suspend animals in M9+gelatin, and then transferred to a 1.5 ml tube. Animals were spun at $700 \times g$ for 1 min. The media was then aspirated off and animals were resuspended in 500 μl M9 + 2% gelatin with 5 mM levamisole. 12 μl of animals in M9 +gel with levamisole solution were placed on slides with a 2% agarose pad and secured with a coverslip. Images were acquired using a Plan-Apochromat 40×/1.3 Oil DIC lens or a Plan-Apochromat 63×/1.4 Oil DIC lens on an AxioImager M2 microscope (Carl Zeiss Microscopy, LLC) equipped with a Colibri 7 LED light source and an Axiocam 506 mono camera. Acquired images were processed through Affinity Photo software (version: 1.9.2.1035). For the confocal microscopy, animals were grown at 25 °C on OP50 plates and picked as L3/L4 stage animals. Worms were mounted on a glass slide with a 2% agarose patch immobilized with 5 μl of 10 mM levamisole (Fluca Analytical, #31742). Images were acquired in channels for red (561 nm laser), green (488 nm laser) and DIC with a 40×/1.3 immersion objective on a Zeiss LSM700 confocal microscope. Acquired images were processed with FiJi (Schindelin et al, 2012).

### Quantification of GFP levels

For quantification, mid-L4 stage worms (defined by vulval morphology) were selected. Images acquired in the GFP fluorescence channel were imported into Napari for analysis (Chiu et al, 2022). The following regions of interest were manually annotated for quantification: Nuclei and cytoplasm of hyp7 and seam cells. For the cytoplasm of hyp7, the layer with the maximum intensity of nuclear GFP signal in hyp7 was selected and a region between the annotated hyp7 nuclei was chosen. To define the background, a region in the gonad was used (where no GFP signal was detected). For the quantification, average GFP levels were computed per region and worm. Background levels were subtracted from the GFP levels of all other regions.

### Microfluidics

Worms were prepared and loaded into the device as described in Berger et al (2021). Briefly, gravid adult hermaphrodites were harvested and treated with bleach. The resulting embryos were collected by centrifugation at $1600 \times g$ for 1 min and washed three times with S-Basal buffer. For synchronization, embryos were incubated overnight in S-Basal, allowing larvae to arrest. Arrested larvae were then passed through a 10 μm filter (pluriStrainer Mini 10 μm, PluriSelect), and 6000 worms were transferred to 3 NGM plates, where they were incubated for 12 h at 25 °C. Once the

worms reached the late L1 stage, they were harvested using M9 buffer, washed twice to remove debris, and loaded into the experimental device. During the experiment, bacterial food was supplied at a constant rate of 1 μl/h, with periodic increases to 100 μl/h for 5 s every 30 min to clear debris. Images were acquired every 10 min over 24 h using a spinning disk confocal scanning microscope (Yokogawa CSU W1 with Dual T2). Brightfield and fluorescent signals (488 nm laser) were recorded simultaneously using two sCMOS Photometrics Prime 95B cameras with a ×40 oil immersion lens (NA = 1.3). Imaging was conducted with a 25 msec exposure time and a motorized z-drive, acquiring z-stacks with a 1 μm step size and 20 images per stack. Analysis was performed using Fiji/ImageJ software.

### Luciferase assays

Assays were performed and analyzed as described (Meeuse et al, 2020). Briefly, gravid adults were bleached, and eggs were immediately singled into wells containing 90 μl OP50/S-Basal/D-Luciferin solution per well and left in the luminometer machine and measured every 10 min for 0.5 s. The assays lasted 96 h or 130 h. For statistical analysis, we performed a Wilcoxon–Mann–Whitney test (implemented in the Python package SciPy version 1.4.1 as the function Mann–Whitney $U$).

### Expression and purification of recombinant proteins

All proteins were expressed from a pET22-based vector in *Escherichia coli* Rosetta (DE3) cells based on the Parallel vector series (Sheffield et al, 1999). LIN-42 C-terminal constructs (CK1BD + Tail, residues 402–598; CK1BD, residues 402–475; CK1BD-ΔA, -ΔB, and ΔAB mutants) were expressed downstream of an N-terminal TEV-cleavable His-NusA tag. LIN-42 N-terminus (residues 1–315) was expressed downstream of a TEV-cleavable HisGβ1tag as before (Lamberti et al, 2024). Human CK1δ catalytic domains (CK1δ ΔC, residues 1–317) were all expressed with a TEV-cleavable His-GST tag. All proteins expressed from Parallel vectors have an additional N-terminal vector artifact of 'GAMD-PEF' remaining after TEV cleavage. Cells were grown in LB media at 37 °C until the O.D.600 reached ~0.8; expression was induced with 0.5 mM IPTG, and cultures were grown for approximately 16–20 h more at 18 °C. Cells were centrifuged at 3200×g, resuspended in 50 mM Tris, pH 7.5, 300 mM NaCl, 20 mM imidazole, 5% (vol/vol) glycerol, 1 mM tris(2-carboxyethyl)phosphine (TCEP), and 0.05% Tween-20. For purification of recombinant protein, cells were lysed with a microfluidizer followed by sonication and then the lysate was clarified via centrifugation at $140,500 \times g$ for 1 h at 4 °C. Ni-NTA affinity chromatography was used to extract his-tagged proteins from the lysate and then the affinity/solubility tags were cleaved using His6-TEV (GST-TEV for CK1δ ΔC) protease overnight at 4 °C. The cleaved protein was then separated from solubility tag and TEV by a second Ni-NTA (GST for CK1δ ΔC) affinity column and further purified using size exclusion chromatography (SEC) in 50 mM Tris, pH 7.5, 200 mM NaCl, 1 mM EDTA, 5% (vol/vol) glycerol, 1 mM TCEP, and 0.05% Tween-20. Small aliquots of protein were frozen in liquid nitrogen and stored at −70 °C for long-term storage.

### In vitro biotinylation, pull-down assays, and bio-layer interferometry

LIN-42 CK1BD + Tail constructs and CK1δ ΔC were biotinylated via Sortase A-mediated reactions between a Sortase A recognition

motif peptide (biotin-LPETGG) and our LIN-42 CK1BD + Tail/CK1δ ΔC (N-terminal G from 'GAMDPEF' artifact). Reactions were carried out in 50 mM Tris pH 7.5 and 150 mM NaCl using 5 µM His6-Sortase A, 300 µM biotin-LPETGG, and 50 µM protein. Ni-NTA affinity chromatography followed by SEC was used to purify labeled protein from His6-Sortase A and excess biotin, respectively. Pull down assays were performed using magnetic streptavidin beads to bind biotinylated LIN-42 CK1BD + Tail WT and mutants (final concentration 5 µM) in the presence and absence of CK1δ ΔC (final concentration 5 µM). All BLI assays were performed in SEC buffer supplemented with 7.5 mM BSA as previously described (Fribourgh et al, 2020; Parico et al, 2020) using an eight-channel Octet-RED96e.

### ADP-Glo kinase assays

Substrate titration kinase reactions were performed on the indicated recombinant LIN-42 proteins (CK1BD + Tail WT or mutants) using the ADP-Glo kinase assay kit (Promega) according to the manufacturer's instructions. All reactions were performed in 30 µL volumes in duplicate ($n = 3$ independent experiments) using 1× kinase buffer (25 mM Tris pH 7.5, 100 mM NaCl, 10 mM MgCl$_2$, and 2 mM TCEP) supplemented with 100 µM ATP, 0.2 µM recombinant CK1, and indicated LIN-42 proteins. Reactions were held at room temperature for 1 h and then 5 µL aliquots were taken and quenched with ADP-Glo reagents. Luminescent measurements were measured at room temperature with a SYNERGY2 microplate reader in 384-well microplates. Data analysis was performed using Excel (Microsoft) and Prism (GraphPad).

### $^{32}$P-ATP kinase assays

1 µM CK1δ ΔC was incubated with 10 µM LIN-42 (CK1BD + Tail/CK1BD WT and mutant proteins) in 1× kinase buffer (25 mM Tris pH 7.5, 100 mM NaCl, 10 mM MgCl2, 2 mM TCEP). Reactions were started by the addition of $^{32}$P-ATP (final concentration 2 mM) and samples were collected at indicated time points and quenched in an equivalent volume of 2× SDS-PAGE loading buffer. Proteins labeled with $^{32}$P were separated and analyzed via SDS-PAGE and the gels were dried at 80 °C for 2 h before overnight exposure in a phosphor screen (Amersham Biosciences). A Typhoon Trio (Amersham Biosciences) phosphorimager was used to visualize exposed gels and $^{32}$P-labeled protein bands were quantified via densitometry using ImageJ (NIH), Excel (Microsoft) and Prism (GraphPad).

### In vitro mass spectrometry

*Sample preparation.*   Kinase reactions were performed in 1× kinase buffer (25 mM Tris pH 7.5, 100 mM NaCl, 10 mM MgCl2, 2 mM TCEP) for 60 min and then quenched with 20 mM EDTA. Samples were denatured, reduced, alkylated, and digested according to the In-solution Protein Digestion (Promega) protocol. In brief, phosphorylated samples were denatured/reduced in 8 M urea/50 mM ammonium bicarbonate/5 mM Dithiothreitol (DTT) for 1 h at 37 °C. Samples were incubated for 30 min in the dark with 15 mM iodoacetamide and then digested with X Grade Modified Trypsin according to the Trypsin Digestion Protocol (Promega). Digested Samples were phospho-enriched using the High-Select Fe-NTA Phosphopeptide Enrichment Kit (Thermo Scientific) and sent for analysis at the University of California, Davis Proteomic Core Facility.

*LC-MS.*   For each sample, equal volumes were loaded onto a disposable Evotip C18 trap column (Evosep Biosystems, Denmark) as per the manufacturer's instructions. Briefly, Evotips were wetted with 2-propanol, equilibrated with 0.1% formic acid, and then loaded using centrifugal force at 1200 × g. Evotips were subsequently washed with 0.1% formic acid, and then 200 µL of 0.1% formic acid was added to each tip to prevent drying. The tipped samples were subjected to nanoLC on a Evosep One instrument (Evosep Biosystems). Tips were eluted directly onto a PepSep analytical column, dimensions: 15 cm × 75 µm C18 column (PepSep, Denmark) with 1.5 µm particle size (100 Å pores) (Bruker Daltonics), and a ZDV spray emitter (Bruker Daltonics). Mobile phases A and B were water with 0.1% formic acid (v/v) and 80/20/0.1% ACN/water/formic acid (v/v/vol), respectively. The standard pre-set method of 40 samples-per-day whisper method was used, which is a 31-min run.

Mass Spectrometry – Performed on a hybrid trapped ion mobility spectrometry-quadrupole time of flight mass spectrometer (timsTOF Pro (Bruker Daltonics, Bremen, Germany) with a modified nano-electrospray ion source (CaptiveSpray, Bruker Daltonics). In the experiments described here, the mass spectrometer was operated in diaPASEF mode. Desolvated ions entered the vacuum region through the glass capillary and deflected into the TIMS tunnel which is electrically separated into two parts (dual TIMS). Here, the first region is operated as an ion accumulation trap that primarily stores all ions entering the mass spectrometer, while the second part performs trapped ion mobility analysis.

*DIA PASEF.*   The dual TIMS analyzer was operated at a fixed duty cycle close to 100% using equal accumulation and ramp times of 85 msec each.

Data-independent analysis (DIA) scheme consisted of one MS scan followed by MSMS scans taken with 19 precursor windows at a width of 50Th per 0.57 s cycle over the mass range 300–1200 Dalton. The TIMS scans layer the doubly and triply charged peptides over a ion mobility −1/k0- range of 0.7–1.3 V*s/cm$^2$. The collision energy was ramped linearly as a function of the mobility from 59 eV at 1/K0 = 1.4 to 20 eV at 1/K0 = 0.6.

*Data analysis.*   DIA: LCMS files were processed with Spectronaut version 16.1 (Biognosys, Zurich, Switzerland) using DirectDIA analysis mode. Mass tolerance/accuracy for precursor and fragment identification was set to default settings. The unreviewed FASTA for *C. elegans* was downloaded from Uniprot and a database of common laboratory contaminants were used (Frankenfield et al, 2022). A maximum of two missing cleavages were allowed, the required minimum peptide sequence length was seven amino acids, and the peptide mass was limited to a maximum of 4600 Da. Carbamidomethylation of cysteine residues was set as a fixed modification, and methionine oxidation and acetylation of protein N-termini as variable modifications. A decoy false discovery rate (FDR) at less than 1% for peptide spectrum matches and protein group identifications was used for spectra filtering (Spectronaut default). Decoy database hits, proteins identified as potential contaminants, and proteins identified exclusively by one site modification were excluded from further analysis.

### Experimental study design and statistics

No statistical estimate of minimal sample size was performed prior to experiments. In the luciferase assays molt annotation occurred semi-automatically and no other additional efforts were made to blind samples. Broods were not counted in animals with high instance of bag-of-worms phenotype.

## Data availability

Data are available from the zenodo repository. https://doi.org/10.5281/zenodo.15719189 (Fig. 6A), https://doi.org/10.5281/zenodo.15723918 (Fig. 6B), https://doi.org/10.5281/zenodo.15799157 (Fig. 6C) and https://doi.org/10.5281/zenodo.15799409 (Fig. 6D). Data are available from zenodo repository, https://doi.org/10.5281/zenodo.15863655 (Fig. EV4). Proteomics Data are available via ProteomeXchange with identifiers PXD058601, PXD058598, and PXD058784. The latter dataset is also available from the massive online repository https://massive.ucsd.edu/, ID number MSV000096641. All other relevant data can be found within the article and its supplementary information.

The source data of this paper are collected in the following database record: biostudies:S-SCDT-10_1038-S44318-025-00585-z.

## Peer review information

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

## Acknowledgements

Some strains were provided by the Caenorhabditis Genetics Center, which is funded by the NIH Office of Research Infrastructure Programs [P40 OD010440]. We thank Iskra Katic for worm injections and technical support, Daniel Hess and Jan Seebacher for mass spectrometry analysis, and Dimos Gaidatzis for help with proteomics data analysis. We are thankful to Laurent Gelman and Laure Plantard for help with confocal imaging and Milou Meeuse for providing plasmid pMM002. Some analysis for this project was performed in the Proteomics Core Facility of the Genome Center, University of California, Davis with instrument funding provided by the NIH (S10OD026918-01A1). We thank Michelle Salemi (UC Davis) for performing the proteomics sample prep, the LC-MS/MS method writing and running and data analysis. The Bruker tims-TOF HT MS/Evosep LC system was supported by the Howard Hughes Medical Institute Investigator Award for Dr. Neal Hunter, UC Davis. Biolayer Interferometry data were collected with instrument funding provided by the NIH (S10OD027012). This work was funded by the National Institutes of Health (NIH) National Institute of General Medical Sciences (NIGMS) to JDW (R01GM138701) and CLP (R35GM141849), and the Howard Hughes Medical Institute to CLP This work is part of a project that has received funding from the European Research Council (ERC) under the European Union's Horizon 2020 research and innovation program (Grant agreement No. 741269, to HG) and from the Swiss National Science Foundation (#310030_207470, to HG). The FMI is core-funded by the Novartis Research Foundation.

## Author contributions

**Rebecca K Spangler**: Formal analysis; Investigation; Writing—original draft. **Kathrin Braun**: Formal analysis; Investigation; Writing—review and editing. **Guinevere E Ashley**: Conceptualization; Formal analysis; Writing—original draft. **Marit van der Does**: Formal analysis; Writing—review and editing. **Daniel Wruck**: Formal analysis; Writing—review and editing. **Andrea Ramos Coronado**: Formal analysis; Writing—review and editing. **James Matthew Ragle**: Formal analysis; Writing—review and editing. **Vytautas Iesmantavicius**: Formal analysis; Writing—review and editing. **Lucas J Morales Moya**: Formal analysis; Writing—review and editing. **Keya Daly**: Formal analysis; Investigation; Writing—review and editing. **Carrie L Partch**: Conceptualization; Supervision; Funding acquisition; Writing—original draft; Project administration. **Helge Großhans**: Conceptualization; Formal analysis; Supervision; Funding acquisition; Writing—original draft. **Jordan D Ward**: Conceptualization; Supervision; Funding acquisition; Writing—original draft; Project administration.

Source data underlying figure panels in this paper may have individual authorship assigned. Where available, figure panel/source data authorship is listed in the following database record: biostudies:S-SCDT-10_1038-S44318-025-00585-z.

## Disclosure and competing interests statement

The authors declare no competing interests.

# Expanded View Figures

**Figure EV1.  Larval stage durations for *lin-42* mutant animals.**

Boxplots showing durations (in hours) for (**A**) larval stages (**B**) molts and (**C**) intermolts from luciferase assay. Wild type in white, *lin-42(n1089)* in dark grey, *lin-42(ok2385)* in light grey, *lin-42(wrd67[ΔPAS])* in blue and *lin-42(wrd63[ΔCK1BD])* in green. Statistics were done using the Mann–Whitney *U* test. Stars indicate the significance of difference between the *Wt* strain and the different *lin-42* mutant animals: *$P < 0.05$, **$P < 0.01$, ***$P < 0.001$, ****$P < 0.0001$. (**D**) Bar plot showing the number of molts from the luciferase assay of the indicated genotypes. Boxplots were generated using the boxplot function in python's seaborn package (v0.13.2) using the default options (center = median, boxes represent values within the 0.25 (Q1) and 0.75 (Q3) quantiles (the interquantile distance or IQR), whiskers represent values within Q1 - 1.5IQR and Q3 + 1.5IQR, and extrema are the minima and maxima for each condition). Two biological replicates were performed for each strain.

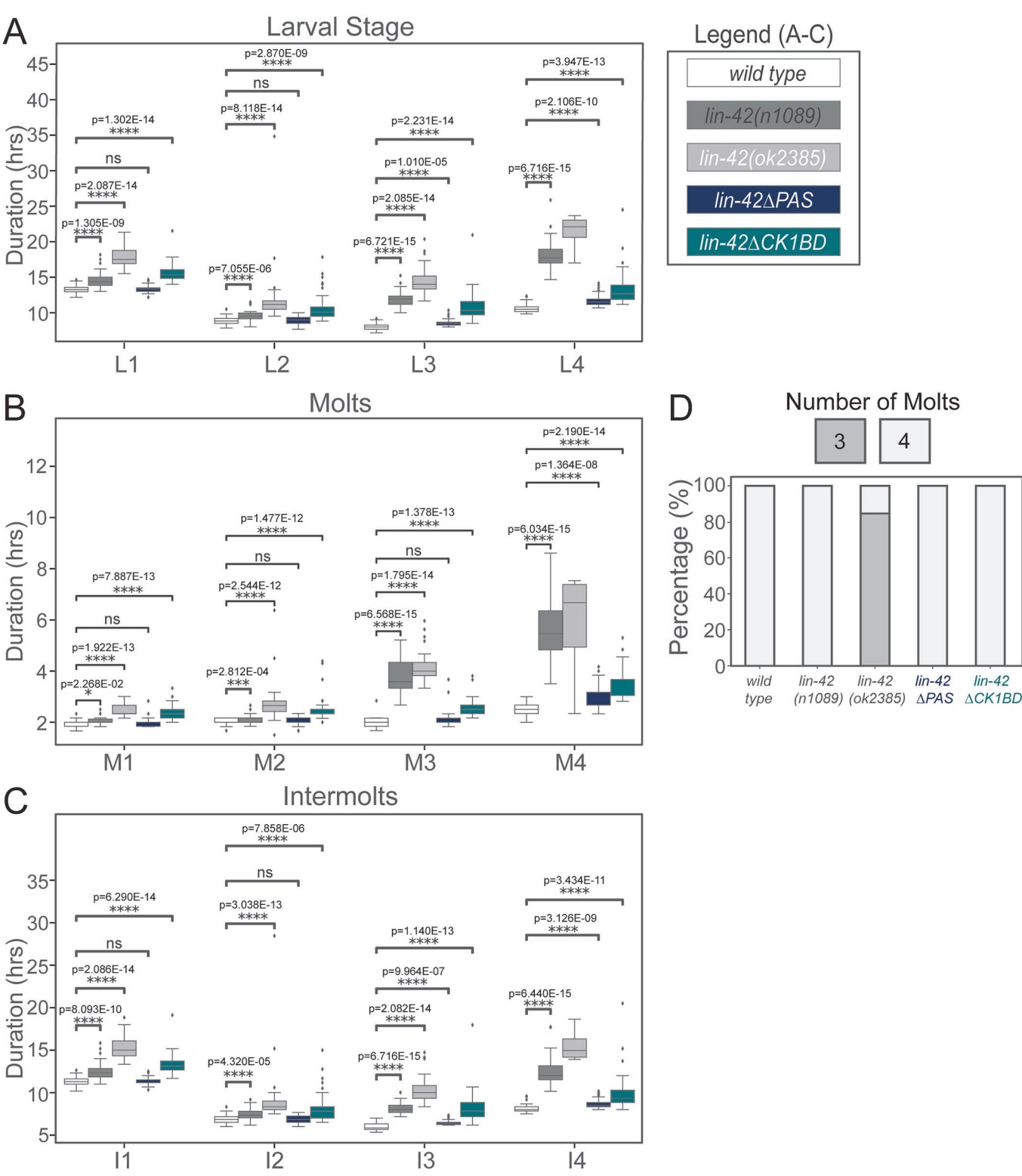

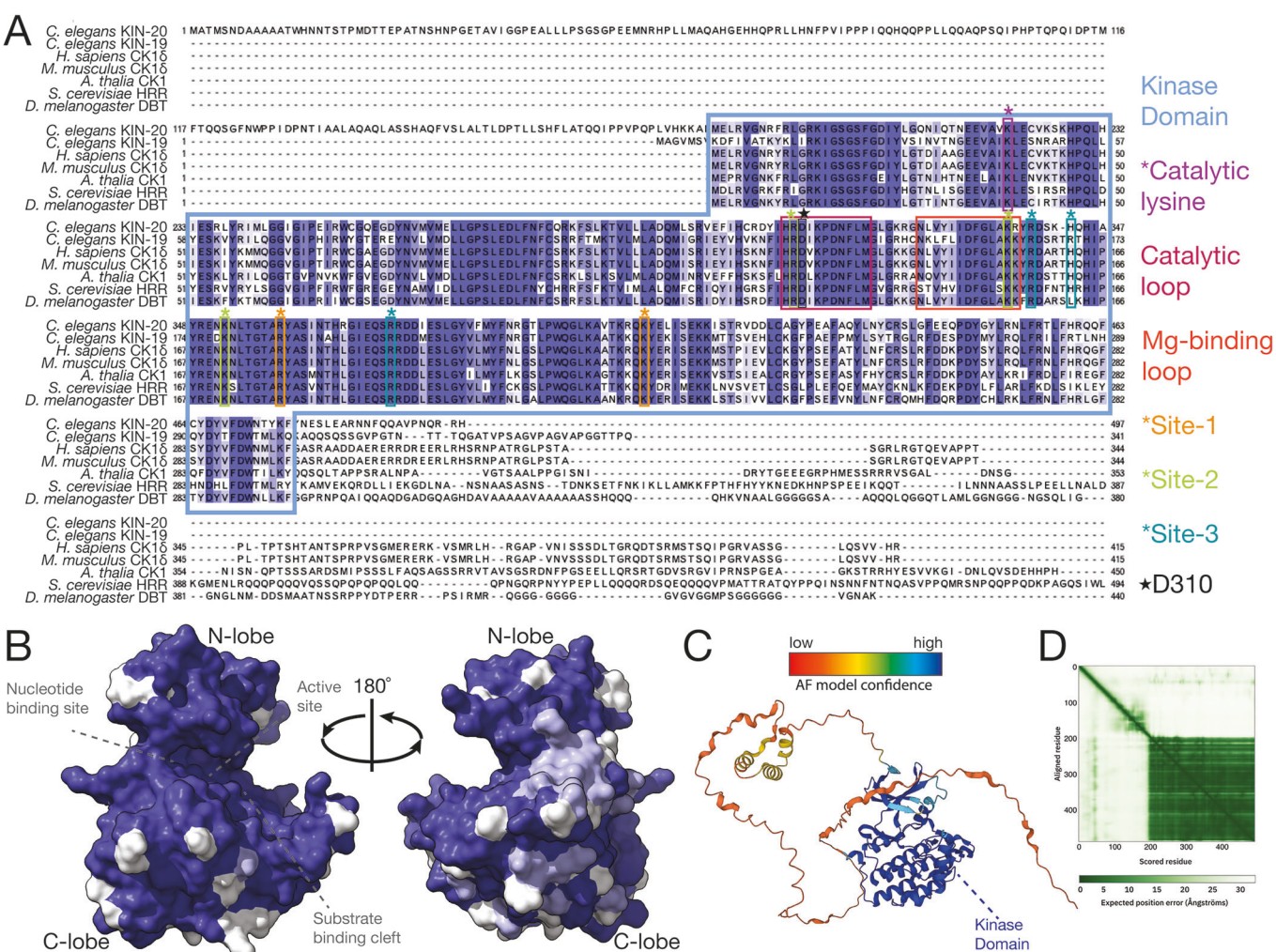

**Figure EV2. Conservation and structural prediction of *C. elegans* KIN-20.**

(A) Sequence alignment of *C. elegans* KIN-20 and KIN-19, *H. sapiens* CK1δ, *M. musculus* CK1δ, *A. thalia* CK1, *S. cerevisiae* HRR25 (HRR), and *D. melanogaster* Doubletime (DBT). Important enzymatic sequence features are boxed; full kinase domain (blue box - 79% identical between KIN-20 and human CK1δ), catalytic lysine (purple - conserved in KIN-20), catalytic loop (pink - 100% conserved between KIN-20 and human CK1δ), Magnesium (Mg)-binding loop (orange - 12 out of 13 residues conserved between KIN-20 and human CK1), anion coordination sites 1 (yellow), 2 (green), and 3 (blue) (all conserved in KIN-20). (B) Crystal structure of *H. sapiens* CK1δ (PDB 6pxo). Dark blue indicates residue is conserved in KIN-20; residues that diverge are highlighted in pale blue (similar amino acid) or white (not conserved). (C) AlphaFold structural prediction of KIN-20 (https://alphafold.ebi.ac.uk/entry/A8X4B3) colored by the model confidence. (D) KIN-20 AlphaFold plot of predicted aligned error.

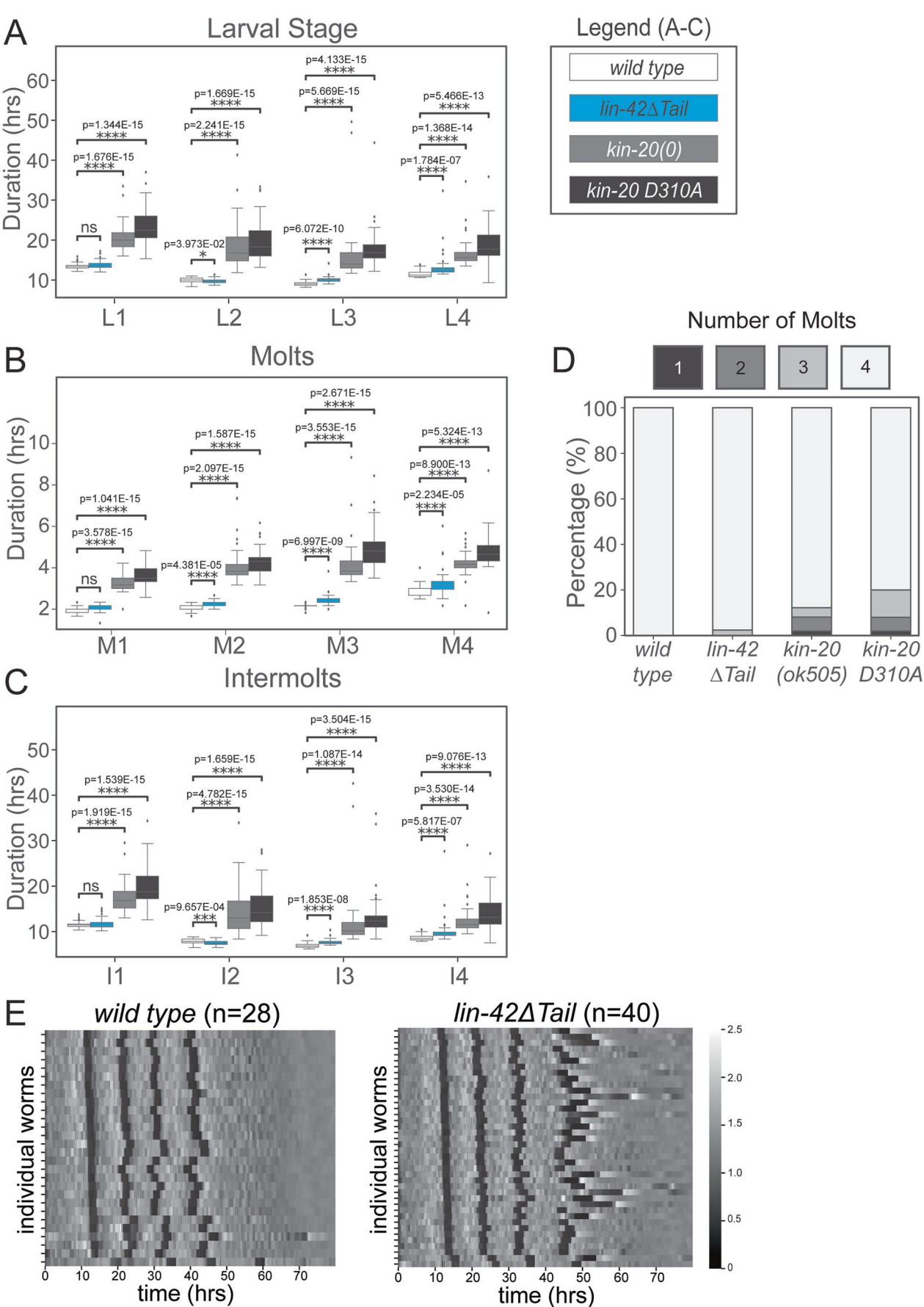

◀ **Figure EV3. Larval stage durations for *kin-20 and lin-42* mutant animals.**

Boxplots showing durations (in hours) for (**A**) larval stages, (**B**) molts and (**C**) intermolts from luciferase assay. Wild type (*Wt*) in white, *lin-42(wrd107[ΔTail])* in blue, *kin-20(0)* in light grey, *kin-20(xe355[D310A])* in dark grey. Statistics were done using the Mann–Whitney *U* test. Stars indicate the significance of difference between the *Wt* strain and the different *lin-42* and *kin-20* mutant animals: *$P < 0.05$, **$P < 0.01$, ***$P < 0.001$, ****$P < 0.0001$. (**D**) Bar plot showing the number of molts detected in the assay in percentage of animals. (**E**) Luciferase replicate of *lin-42(wrd107[ΔTail]*. Heatmaps showing trend-corrected luminescence traces from the indicated genotype. Each horizontal line represents one animal. Traces are sorted to the entry of the first molt. Darker color indicates low luminescence signal and corresponds to the molts. Boxplots were generated using the boxplot function in Python's seaborn package (v0.13.2) using the default options (center = median, boxes represent values within the 0.25 (Q1) and 0.75 (Q3) quantiles (the interquantile distance or IQR), whiskers represent values within Q1 - 1.5IQR and Q3 + 1.5IQR, and extrema are the minima and maxima for each condition). Two biological replicates were performed for each strain. Source data are available online for this figure.

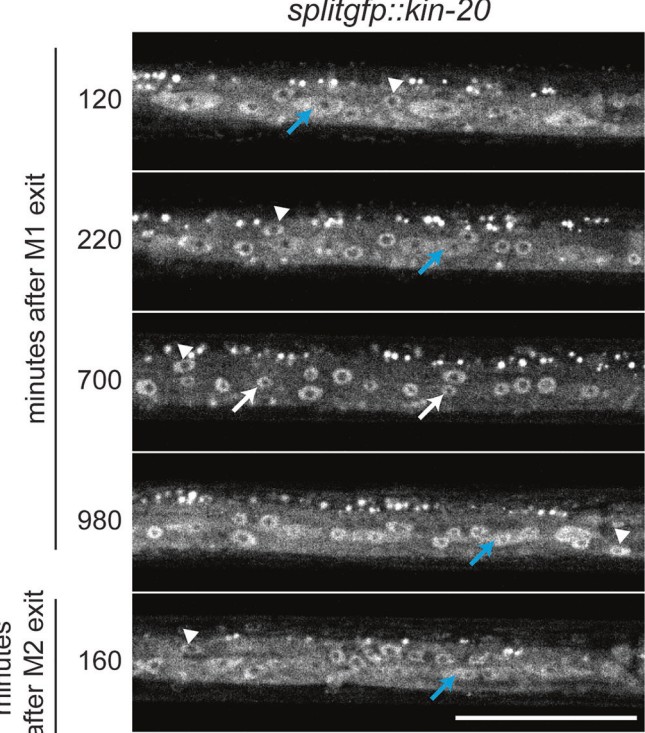

**Figure EV4. KIN-20 dynamics during L2 - L3 stage.**

Time-lapse microscopy images of a single *splitgfp::kin-20* larva (*bchSi84[eft-3p::gfp1-10(codon-optimized)::tbb-2 3UTR]); kin-20[(xe354)[3xflag::4xgfp11]])* followed in a microfluidics device over time. Time indicated in minutes after molt 1 (M1) or molt 2 (M2) exit. Arrows indicate seam cell nuclear (white) and cytoplasmic (blue) localization; white arrowheads indicate nuclear hyp7 localization. Scale bar = 50 μm. This experiment was performed once. Source data are available online for this figure.

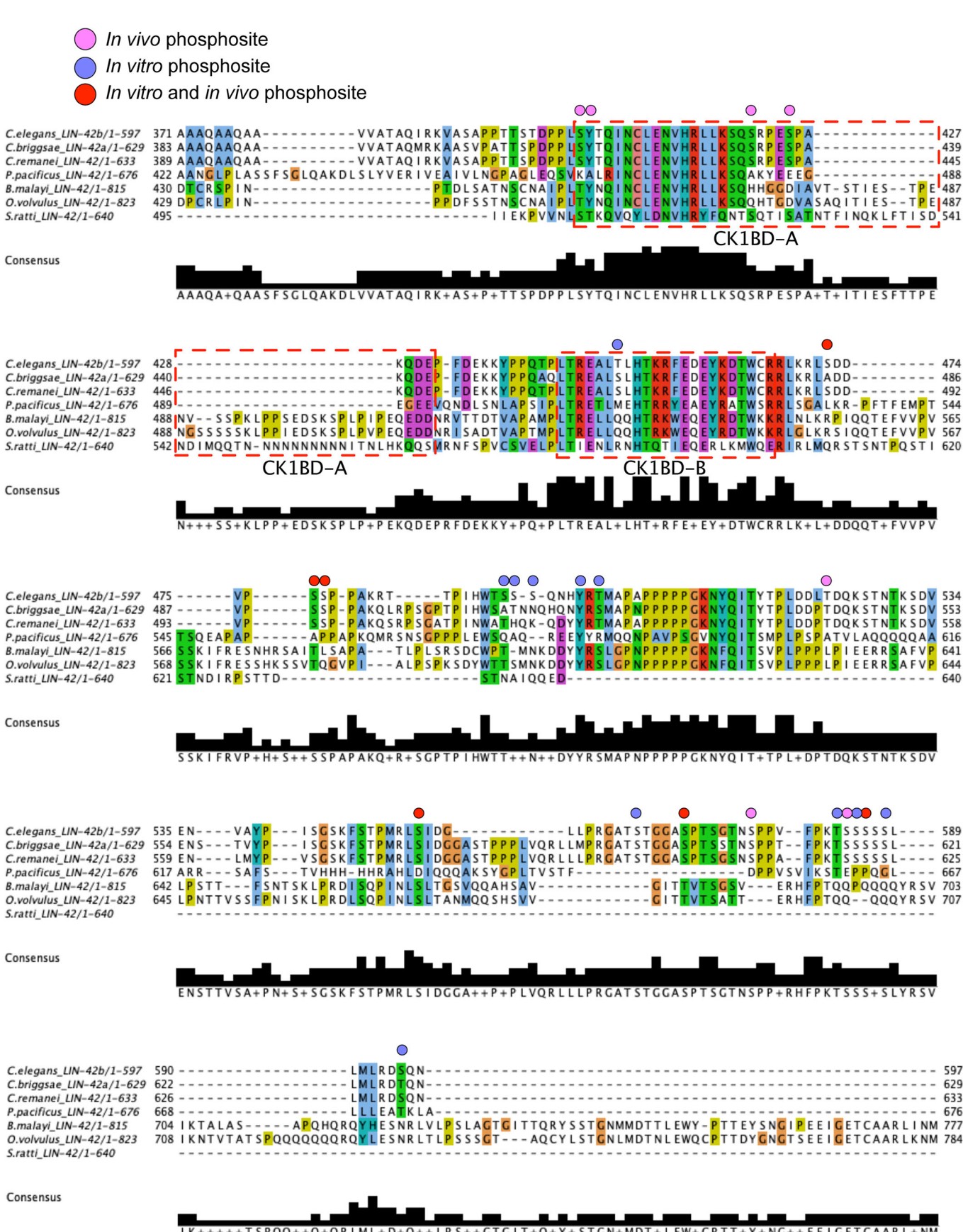

**Figure EV5.   Alignment of nematode LIN-42 protein sequences.**

LIN-42 homologs from the indicated nematode species were aligned using Clustal Omega. The length in amino acids of each homolog follows the species and homolog name. To the left and right of the alignment are amino acid positions of the end residues for each protein. Blue shading indicates conserved sequences and the histogram at the bottom depicts the degree of conservation with a consensus sequence listed below. The positions of the *C. elegans* CK1BD-A and CK1BD-B motifs are indicated. The location of the phosphosites found in our in vivo, in vitro and both datasets are indicated.

