## [Peer Review File · The EMBO Journal]

A conserved chronobiological complex times *C. elegans* development

Rebecca Spangler, Kathrin Braun, Guinevere Ashley, Marit van der Does, Daniel Wruck, Andrea Ramos Coronado, James Ragle, Vytautas Iesmantavicius, Lucas Morales Moya, Keya Daly, Carrie Partch, Helge Großhans, and Jordan Ward

Corresponding authors: Jordan Ward (jward2@ucsc.edu) , Helge Großhans (helge.grosshans@fmi.ch)

Review Timeline:

Transferred from Review Commons:	13th Mar 25
Editorial Decision:	18th Mar 25
Revision Received:	18th Jul 25
Editorial Decision:	27th Aug 25
Revision Received:	24th Sep 25
Accepted:	25th Sep 25

Editor: Ieva Gailite

Transaction Report:

Review #1**1. Evidence, reproducibility and clarity:****Evidence, reproducibility and clarity (Required)**

The authors investigate in this study the function of LIN-42 for the process of precise molting timing in *C. elegans*. To achieve this, they compare LIN-42 with its mammalian ortholog, Period. They found that similar to Period, LIN-42 interacted with the kinase KIN-20, a mammalian Casein kinase 1 (CK1) ortholog. Hence, two different proteins involved in rhythmic processes, LIN-42 and Period function in a conserved manner.

First, they used mutants with specific deletions to untangle various phenotypes during *C. elegans* development. From this analysis they identify a specific region, corresponding to a CK1-binding region in mammals, to be mainly involved in the rhythmic molting phenotype. Next, they identify KIN-20, the CK1 ortholog as interaction partner of LIN-42. They even were able to demonstrate an interaction of CK1 with the region of LIN-42. Using CK1, they identified potential phosphorylation sites within LIN-42 and compared those with immunoprecipitated protein *in vivo*. There was a substantial overlap. While the C-terminal tail of LIN-42 was heavily phosphorylated, deletion of the C-terminal part resulted only in a minor phenotype for rhythmic molting. Last but not least, they demonstrated that point mutations that inactivate the catalytic function of KIN-20 produced a rhythmic molting phenotype. The interaction of LIN-42 with KIN-20 affected the localization of the kinase, similar to what was found to Period and CK1.

Overall, the experiments are well done, well controlled and well described even for non-specialists. I guess it was not easy to kind of sort out the many overlapping phenotypes. It was certainly helpful just to focus on the clear rhythmic molting phenotype.

I have no major or minor comments.

2. Significance:**Significance (Required)**

The manuscript is well written and can be followed by non-specialists of the field. The experiments are well performed. Even if some experiments did not yield the expected phenotype, e.g. deletion of the C-terminal tail of LIN-42 had only a minor phenotype inspire of heavy phosphorylation, these experiments are anyhow included and explained. Overall, the study is interesting for people in the *C. elegans* field and by similarity mammalian chronobiology. I would expect that most of the progress based on this study

will be on the further elucidation of the molting phenotype and how the other phenotypes related to this. Then this could emerge as a blueprint for molting phenomena in other species as well.

I am a mammalian chronobiologist working on Period proteins.

3. How much time do you estimate the authors will need to complete the suggested revisions:

Estimated time to Complete Revisions (Required)

(Decision Recommendation)

Less than 1 month

4. Review Commons values the work of reviewers and encourages them to get credit for their work. Select 'Yes' below to register your reviewing activity at Web of Science Reviewer Recognition Service (formerly Publons); note that the content of your review will not be visible on Web of Science.

No

Review #2

1. Evidence, reproducibility and clarity:

Evidence, reproducibility and clarity (Required)

This study represents pioneering work on LIN-42, the *C. elegans* ortholog of PER, uncovering its role in molting rhythms and heterochronic timing. A key strength of this work lies in its integrative approach, combining genetic and developmental analyses in *C. elegans* with biochemical characterization of LIN-42 protein.

At the organismal level, the authors take advantage of the power of *C. elegans* as a model system, employing precise genetic manipulations and high-resolution developmental assays to dissect the contributions of LIN-42 and its interaction partner KIN-20, the *C. elegans* ortholog of CK1, to molting rhythms. Their findings provide *in vivo* evidence that binding of LIN-42 with KIN-20 promotes the nuclear accumulation of KIN-20 and is crucial for molting rhythms, while its PAS domain appears dispensable for this function. This detailed phenotypic analysis of multiple LIN-42 and KIN-20 mutants represents a

significant contribution to our understanding of the developmental clock.

At the biochemical level, the study provides a detailed analysis of the mechanism underlying LIN-42's interaction with CK1, demonstrating that LIN-42 contains a functionally conserved CK1-binding domain (CK1BD). Through their in vitro kinase assays and structural insights, the authors identified distinct roles for CK1BD-A and CK1BD-B: the former in kinase inhibition and the latter in stable CK1 binding and phosphorylation. Importantly, their data align well with previous findings on PER-CK1 regulation in mammalian and *Drosophila* systems, reinforcing the evolutionary conservation of key clock components.

Overall, this work stands out for its deep and important insights into how CK1-mediated regulation extends beyond the circadian clock to regulate the developmental clock. The combination of genetic approaches with biochemical analyses makes this an outstanding contribution to both chronobiology and nematode developmental biology.

Major comment 1:

In Figure 2D, I could not find a crucial control if the authors claim that KIN-20 binds to LIN-42. For example, a single mutant of LIN-42-3xFLAG could be used as a control for the double mutant.

Major comment 2:

The sizes of the KIN20 bands were very diverged (~40 kDa and ~60 kDa), but the authors provide no explanation for this.

Major comment 3:

Regarding the MS study, the raw data are available, but the detailed supplemental Excel files would be more informative for readers. For example, are other interactors such as REV-ERB/NHR-85 detected in Figure 2A? Regarding Figure 4F, the list of phosphorylation sites and MS scores is also informative.

Major comment 4:

It is an important finding that the PAS domain of LIN-42 is not essential for the molting

rhythms. Is the PAS domain also dispensable for binding with KIN-20?

Major comment 5 (Optional):

In this study, the authors carefully performed in vitro kinase assays, and I strongly suggest that they investigate whether the CKI-mediated phosphorylation of LIN-42 is temperature-compensated and whether the CKI-BD-AB regions affect it.

Major comment 6 (Optional):

In Figure 6, the authors argue that the CKI-BD of LIN-42 is important for CK1 nuclear translocation. It would be better to show the effect of the nuclear accumulation of CKI on nuclear proteins, like the mammalian CKI-PER2-CLOCK story. Does CKI localization affect phosphorylation status of other clock-related proteins including REV-ERB/NHR-85? Phospho-proteome analysis would identify nuclear substrates of CK1. In addition, is phosphorylation of LIN-42 dispensable for the CK1 nuclear translocation?

Major comment 7 (Optional):

LIN-42 rhythmic expression could drive rhythmic nuclear accumulation of KIN-20. It would be better to examine this possibility using kin-20::GFP in lin-42 mutants.

Minor 1:

I could not find the full gel images of the Western blot analyses as supplemental materials.

Minor 2:

The authors discussed a conserved module in two different clocks. A statement regarding a recently published paper (Hiroki and Yoshitane, Commun Biol, 2024) would be informative for readers.

Referee cross-commenting

I basically agree with reviewer 1 and hope that this paper will be published soon as it is very valuable for our field. I have constructively pointed out some parts that could be improved,

but depending on the editor's judgement, I believe that even if not all of these are revised, it will be sufficient for publication.

2. Significance:

Significance (Required)

This work stands out for its deep and important insights into how CK1-mediated regulation extends beyond the circadian clock to regulate the developmental clock. The combination of genetic approaches with biochemical analyses makes this an outstanding contribution to both chronobiology and nematode developmental biology.

I strongly suggest editors to accept this study with minor modifications according to the following comments.

3. How much time do you estimate the authors will need to complete the suggested revisions:

Estimated time to Complete Revisions (Required)

(Decision Recommendation)

Between 1 and 3 months

4. Review Commons values the work of reviewers and encourages them to get credit for their work. Select 'Yes' below to register your reviewing activity at Web of Science Reviewer Recognition Service (formerly Publons); note that the content of your review will not be visible on Web of Science.

Yes

Review #3

1. Evidence, reproducibility and clarity:

Evidence, reproducibility and clarity (Required)

In their manuscript "A conserved chronobiological complex times *C. elegans* development", Spangler, Braun, Ashley et al. investigate the mechanisms through which the PERIOD orthologue, *lin-42*, regulates rhythmic molting in *C. elegans*. Through precise genetic manipulations, the authors identify a particular region of *lin-42*, the 'CK1BD', which regulates molting timing, with less effect on other *lin-42* phenotypes (e.g. heterochrony).

They show that LIN-42 and the casein kinase 1 (CK1) homologue KIN-20 interact in vivo, and identify phosphorylation sites of LIN-42. Using biochemical assays, they find that the CK1BD of LIN-42 is sufficient for interaction with the human homologue of KIN-20, CK1, in vitro. The LIN-42 CK1BD is also required for the proper nuclear accumulation of KIN-20 in vivo. Furthermore, a point mutation that should disrupt the catalytic activity of KIN-20 also shows an irregular molting phenotype, similar to the lin-42 CK1BD mutant. The manuscript is very well-written and the data and methods are well-presented and detailed. Overall this work makes a convincing case that the *C. elegans* lin-42:Kin-20 and mammalian period:Ck1 interactions have functionally conserved roles in the oscillatory developmental programs of each organism (molting timing and circadian rhythms, respectively), with a few caveats below that can be addressed.

****Major comments:****

1. The authors have shown that LIN-42 is phosphorylated in vivo, but the dependence of this phosphorylation on KIN-20 is not fully addressed. In the discussion (lines 417-420), the authors mention that the unhealthy phenotype of the kin-20 mutant animals prevented them from assessing LIN-42 phosphorylation in this genetic background. To bolster their model and to circumvent this issue, it should be feasible to generate a kin-20 degron allele and to perform the LIN-42 phospho-proteomics upon inducible degradation. Alternatively, perhaps a phos-tag western blot for LIN-42 could be used to compare the kin-20 wild-type to kin-20 mutants.

2. For technical reasons, the in vitro biochemistry was done using human CK1 protein. There are a few places (e.g. results, line 248 and discussion line 437), where the language, in my opinion, is extrapolating the CK1 results too strongly to KIN-20. The authors mention that feedback inhibition is a known property of human CK1. It is indeed quite striking that the LIN-42 CK1BD region interacts with and is phosphorylated by the human counterpart of KIN-20, and that feedback inhibition is also seen! However, the language about KIN-20 itself should be softened, since there does not appear to be clear evidence that KIN-20 exhibits the same properties as human CK1 (unless perhaps human CK1 can functionally replace KIN-20 in worms, or the proteins were extremely similar?)

3. The role of the three LIN-42 isoforms should be further clarified. Minimally, it should be explained why the alleles where both b and c isoforms should be flag-tagged seem to only produce detectable b isoform (e.g. Fig. 2C).

4. Related to points 2 and 3 above, the authors have shown that the CK1BD mediates association with human CK1 in vitro, and is required for nuclear accumulation of KIN-20 in vivo, but not that the complex formation between LIN-42 and KIN-20 depends on the CK1BD. Given the reciprocal co-IP findings, it should be feasible to create tagged versions

of lin-42(deltaCK1BD) and to determine the effect on LIN-42-KIN-20 complex formation. While there is already a b-isoform tag, an a-isoform tag would also help to address whether both the b and a isoforms interact with KIN-20 in a CK1BD-dependent manner in vivo. These strains would also allow the authors to determine how the CK1BD deletion affects overall levels/stability/rhythmic accumulation of LIN-42(a or b), which would potentially serve to strengthen their conclusions about the role of the lin-42 CK1BD.

5. In the molting timing assay, there is an unexpected result where the delta-C-terminal-tail lin-42 allele resembles the n1089 (N-terminal deletion) (line 315). Could the authors more clearly explain this finding?

****Minor comments:****

1. The correspondence between the LIN-42 "SYQ" and "LT" motifs and the motifs referred to as "A" and "B" should be clarified, and consistent names/labels used. Are these interchangeable names? If it is necessary to use both names, the differences between SYQ/LT and A/B should be made more clear.
2. For data presented as "% of animals", please indicate the number of animals scored (e.g. egl, alae assays - ~ how many animals per replicate (dot)?).
3. Line 145-148 - Mentioning the relevant phenotype(s) of the lin-42 null allele from the cited paper would provide a good point of comparison here.
4. Line 201 - the phrase "This is also true for the proteins:" is unclear, as the previous sentence states that both lin-42 and kin-20 mRNAs oscillate, while the next sentence says that only LIN-42 protein oscillates.
5. Line 231 - please explain the significance of the 'lower response signal' in the BLI assay for the CK1BD(no tail).
6. Fig. 2 - C/D - the genotype lane labels should I think indicate an N-terminal rather than C-terminal LIN-42 tag.
7. Fig. 6, line 367 - lin-42 is variably described as promoting increased KIN-20 'nuclear accumulation' or 'localization'. I think that 'accumulation' is more accurate, as it doesn't imply a specific mechanism for the difference (transport vs stabilization, etc.)
8. Fig 6B - an overlay of the panels or another way of quantifying the colocalization would make this result more clear.

2. Significance:

Significance (Required)

This work presents a major mechanistic and conceptual advance in our understanding of the role of lin-42/Period, a conserved key regulator of *C. elegans* development. Previously,

it was not clear if the heterochronic and circadian functions of lin-42 were genetically separable, nor was it known how LIN-42 physically interacted with the CK1 homologue. This work addresses these questions using precise genome engineering and detailed phenotypic and biochemical approaches. The work also reveals the conservation of bi-directional/reciprocal regulation between lin-42 and kin-20. The main limitations of the study, which can potentially be addressed as outlined in the 'major points' above, are that evidence should be provided that lin-42 phosphorylation depends on kin-20 in vivo, and that the CK1BD mediates the interaction in vivo (since the in vitro work is with human CK1). As the authors indicate, this is the first 'conserved clock module' of this type, and this work will therefore be of significant interest to both the C. elegans developmental biology and the more general biological timing fields.

Field of expertise of the reviewer- C. elegans genetics and development.

3. How much time do you estimate the authors will need to complete the suggested revisions:

Estimated time to Complete Revisions (Required)

(Decision Recommendation)

Between 1 and 3 months

4. Review Commons values the work of reviewers and encourages them to get credit for their work. Select 'Yes' below to register your reviewing activity at Web of Science Reviewer Recognition Service (formerly Publons); note that the content of your review will not be visible on Web of Science.

No

Revision Plan

Manuscript number: RC-2024-02838

Corresponding author(s): Jordan Ward & Helge Großhans

[The "revision plan" should delineate the revisions that authors intend to carry out in response to the points raised by the referees. It also provides the authors with the opportunity to explain their view of the paper and of the referee reports.]

The document is important for the editors of affiliate journals when they make a first decision on the transferred manuscript. It will also be useful to readers of the reprint and help them to obtain a balanced view of the paper.

*If you wish to submit a full revision, please use our "Full Revision" template. **It is important to use the appropriate template to clearly inform the editors of your intentions.**]*

1. General Statements [optional]

We thank the reviewers for their insightful and positive reviews.

2. Description of the planned revisions

Reviewer #1 (Evidence, reproducibility and clarity (Required)):

The authors investigate in this study the function of LIN-42 for the process of precise molting timing in *C. elegans*. To achieve this, they compare LIN-42 with its mammalian ortholog, Period. They found that similar to Period, LIN-42 interacted with the kinase KIN-20, a mammalian Casein kinase 1 (CK1) ortholog. Hence, two different proteins involved in rhythmic processes, LIN-42 and Period function in a conserved manner.

First, they used mutants with specific deletions to untangle various phenotypes during *C. elegans* development. From this analysis they identify a specific region, corresponding to a CK1-binding region in mammals, to be mainly involved in the rhythmic molting phenotype. Next, they identify KIN-20, the CK1 ortholog as interaction partner of LIN-42. They even were able to demonstrate an interaction of CK1 with the region of LIN-42. Using CK1, they identified potential phosphorylation sites within LIN-42 and compared those with immunoprecipitated protein *in vivo*. There was a substantial overlap. While the C-terminal tail of LIN-42 was heavily phosphorylated, deletion of the C-terminal part resulted only in a minor phenotype for rhythmic molting. Last but not least, they demonstrated that point mutations that inactivate the catalytic function of KIN-20 produced a rhythmic molting phenotype. The interaction of LIN-42 with KIN-20 affected the localization of the kinase, similar to what was found to Period and CK1.

Overall, the experiments are well done, well controlled and well described even for non-specialists. I guess it was not easy to kind of sort out the many overlapping phenotypes. It

Revision Plan

was certainly helpful just to focus on the clear rhythmic molting phenotype.

I have no major or minor comments.

Reviewer #1 (Significance (Required)):

The manuscript is well written and can be followed by non-specialists of the field. The experiments are well performed. Even if some experiments did not yield the expected phenotype, e.g. deletion of the C-terminal tail of LIN-42 had only a minor phenotype inspire of heavy phosphorylation, these experiments are anyhow included and explained. Overall, the study is interesting for people in the *C. elegans* field and by similarity mammalian chronobiology. I would expect that most of the progress based on this study will be on the further elucidation of the molting phenotype and how the other phenotypes related to this. Then this could emerge as a blueprint for molting phenomena in other species as well.

I am a mammalian chronobiologist working on Period proteins.

We thank the reviewer for their positive evaluation of our work.

Reviewer #2 (Evidence, reproducibility and clarity (Required)):

This study represents pioneering work on LIN-42, the *C. elegans* ortholog of PER, uncovering its role in molting rhythms and heterochronic timing. A key strength of this work lies in its integrative approach, combining genetic and developmental analyses in *C. elegans* with biochemical characterization of LIN-42 protein.

At the organismal level, the authors take advantage of the power of *C. elegans* as a model system, employing precise genetic manipulations and high-resolution developmental assays to dissect the contributions of LIN-42 and its interaction partner KIN-20, the *C. elegans* ortholog of CK1, to molting rhythms. Their findings provide *in vivo* evidence that binding of LIN-42 with KIN-20 promotes the nuclear accumulation of KIN-20 and is crucial for molting rhythms, while its PAS domain appears dispensable for this function. This detailed phenotypic analysis of multiple LIN-42 and KIN-20 mutants represents a significant contribution to our understanding of the developmental clock.

At the biochemical level, the study provides a detailed analysis of the mechanism underlying LIN-42's interaction with CK1, demonstrating that LIN-42 contains a functionally conserved CK1-binding domain (CK1BD). Through their *in vitro* kinase assays and structural insights, the authors identified distinct roles for CK1BD-A and CK1BD-B: the former in kinase inhibition and the latter in stable CK1 binding and phosphorylation. Importantly, their data align well with previous findings on PER-CK1 regulation in mammalian and *Drosophila* systems, reinforcing the evolutionary conservation of key clock components.

Overall, this work stands out for its deep and important insights into how CK1-mediated regulation extends beyond the circadian clock to regulate the developmental clock. The

Revision Plan

combination of genetic approaches with biochemical analyses makes this an outstanding contribution to both chronobiology and nematode developmental biology.

We thank the reviewer for the strong endorsement for publication of our work

Major comment 1:

In Figure 2D, I could not find a crucial control if the authors claim that KIN-20 binds to LIN-42. For example, a single mutant of LIN-42-3xFLAG could be used as a control for the double mutant.

We will do an appropriate control experiment.

Major comment 2:

The sizes of the KIN20 bands were very diverged (~40 kDa and ~60 kDa), but the authors provide no explanation for this.

The worm produces several KIN-20 isoforms. We will state this in the revised manuscript.

Major comment 3:

Regarding the MS study, the raw data are available, but the detailed supplemental Excel files would be more informative for readers. For example, are other interactors such as REV-ERB/NHR-85 detected in Figure 2A? Regarding Figure 4F, the list of phosphorylation sites and MS scores is also informative.

We apologize for our omission in stating clearly in the figure legend that the significantly enriched proteins were labeled with a red dot. These were only LIN-42 itself and KIN-20. NHR-85 was not enriched. We will state this explicitly in a revised version and provide all relevant information.

Major comment 4:

It is an important finding that the PAS domain of LIN-42 is not essential for the molting rhythms. Is the PAS domain also dispensable for binding with KIN-20?

Although we have currently no reason to assume that the PAS domain would be required for KIN-20 binding, we will perform an in vitro experiment to test for binding.

Major comment 5 (Optional):

In this study, the authors carefully performed in vitro kinase assays, and I strongly suggest that they investigate whether the CKI-mediated phosphorylation of LIN-42 is temperature-compensated and whether the CKI-BD-AB regions affect it.

Although this is an interesting question, addressing it appears outside the scope of the manuscript and a revision; please see section 4 below.

Major comment 6 (Optional):

In Figure 6, the authors argue that the CKI-BD of LIN-42 is important for CK1 nuclear

Revision Plan

translocation. It would be better to show the effect of the nuclear accumulation of CKI on nuclear proteins, like the mammalian CKI-PER2-CLOCK story. Does CKI localization affect phosphorylation status of other clock-related proteins including REV-ERB/NHR-85? Phospho-proteome analysis would identify nuclear substrates of CK1. In addition, is phosphorylation of LIN-42 dispensable for the CK1 nuclear translocation?

This is another interesting question yet currently nothing is known about other CK1/KIN-20 targets, and we have no evidence for NHR-85 being one. Please see our detailed comments in the section 4 below.

To address whether LIN-42 phosphorylation affects CK1/KIN-20 nuclear accumulation, we will seek to examine KIN-20 localization in LIN-42 Δ Tail animals.

Major comment 7 (Optional):

LIN-42 rhythmic expression could drive rhythmic nuclear accumulation of KIN-20. It would be better to examine this possibility using kin-20::GFP in lin-42 mutants.

We agree that the mutant analysis is important for this and Fig. 6C shows reduced KIN-20 nuclear accumulation in LIN-42 Δ CK1BD.

Minor 1:

I could not find the full gel images of the Western blot analyses as supplemental materials.

This data will be added.

Minor 2:

The authors discussed a conserved module in two different clocks. A statement regarding a recently published paper (Hiroki and Yoshitane, Commun Biol, 2024) would be informative for readers.

We will add such a statement.

Referee cross-commenting

I basically agree with reviewer 1 and hope that this paper will be published soon as it is very valuable for our field. I have constructively pointed out some parts that could be improved, but depending on the editor's judgement, I believe that even if not all of these are revised, it will be sufficient for publication.

Reviewer #2 (Significance (Required)):

This work stands out for its deep and important insights into how CK1-mediated regulation extends beyond the circadian clock to regulate the developmental clock. The combination of genetic approaches with biochemical analyses makes this an outstanding contribution to both chronobiology and nematode developmental biology.

Revision Plan

I strongly suggest editors to accept this study with minor modifications according to the following comments.

We thank the reviewer for their strong support and the clear indication of required vs. optional revisions.

Reviewer #3 (Evidence, reproducibility and clarity (Required)):

In their manuscript "A conserved chronobiological complex times *C. elegans* development", Spangler, Braun, Ashley et al. investigate the mechanisms through which the PERIOD orthologue, *lin-42*, regulates rhythmic molting in *C. elegans*. Through precise genetic manipulations, the authors identify a particular region of *lin-42*, the 'CK1BD', which regulates molting timing, with less effect on other *lin-42* phenotypes (e.g. heterochrony). They show that LIN-42 and the casein kinase 1 (CK1) homologue KIN-20 interact in vivo, and identify phosphorylation sites of LIN-42. Using biochemical assays, they find that the CK1BD of LIN-42 is sufficient for interaction with the human homologue of KIN-20, CK1, in vitro. The LIN-42 CK1BD is also required for the proper nuclear accumulation of KIN-20 in vivo. Furthermore, a point mutation that should disrupt the catalytic activity of KIN-20 also shows an irregular molting phenotype, similar to the *lin-42* CK1BD mutant. The manuscript is very well-written and the data and methods are well-presented and detailed. Overall this work makes a convincing case that the *C. elegans* *lin-42*:Kin-20 and mammalian period:Ck1 interactions have functionally conserved roles in the oscillatory developmental programs of each organism (molting timing and circadian rhythms, respectively), with a few caveats below that can be addressed.

We thank the reviewer for their positive evaluation of our work.

Major comments:

1. The authors have shown that LIN-42 is phosphorylated in vivo, but the dependence of this phosphorylation on KIN-20 is not fully addressed. In the discussion (lines 417-420), the authors mention that the unhealthy phenotype of the *kin-20* mutant animals prevented them from assessing LIN-42 phosphorylation in this genetic background. To bolster their model and to circumvent this issue, it should be feasible to generate a *kin-20* degron allele and to perform the LIN-42 phospho-proteomics upon inducible degradation. Alternatively, perhaps a phos-tag western blot for LIN-42 could be used to compare the *kin-20* wild-type to *kin-20* mutants.

We agree, and acknowledged in the discussion, that phosphorylation of LIN-42 by KIN-20 in vivo has not been demonstrated by us. However, as discussed in the section 4 below, we find that this costly, challenging and time-consuming experiment is not warranted by the expected gain.

2. For technical reasons, the in vitro biochemistry was done using human CK1 protein.

Revision Plan

There are a few places (e.g. results, line 248 and discussion line 437), where the language, in my opinion, is extrapolating the CK1 results too strongly to KIN-20. The authors mention that feedback inhibition is a known property of human CK1. It is indeed quite striking that the LIN-42 CK1BD region interacts with and is phosphorylated by the human counterpart of KIN-20, and that feedback inhibition is also seen! However, the language about KIN-20 itself should be softened, since there does not appear to be clear evidence that KIN-20 exhibits the same properties as human CK1 (unless perhaps human CK1 can functionally replace KIN-20 in worms, or the proteins were extremely similar?)

We will follow the reviewer's advice and carefully examine the text for instances where we extrapolated too much and tone these down. (We note that this does not apply to the example of line 248 where we wrote "Collectively, our data establish that the LIN-42 CK1BD is functionally conserved and mediates stable binding to the CK1 kinase domain.", i.e., there was no mentioning of KIN-20.)

3. The role of the three LIN-42 isoforms should be further clarified. Minimally, it should be explained why the alleles where both b and c isoforms should be flag-tagged seem to only produce detectable b isoform (e.g. Fig. 2C).

We will clarify that the individual roles of the isoforms are largely unknown and that we can only speculate that the c-isoform may exhibit either generally low expression or expression in only few cells or tissues.

4. Related to points 2 and 3 above, the authors have shown that the CK1BD mediates association with human CK1 in vitro, and is required for nuclear accumulation of KIN-20 in vivo, but not that the complex formation between LIN-42 and KIN-20 depends on the CK1BD. Given the reciprocal co-IP findings, it should be feasible to create tagged versions of lin-42(Δ CK1BD) and to determine the effect on LIN-42-KIN-20 complex formation. While there is already a b-isoform tag, an a-isoform tag would also help to address whether both the b and a isoforms interact with KIN-20 in a CK1BD-dependent manner in vivo. These strains would also allow the authors to determine how the CK1BD deletion affects overall levels/stability/rhythmic accumulation of LIN-42(a or b), which would potentially serve to strengthen their conclusions about the role of the lin-42 CK1BD.

We will attempt to generate a FLAG-tagged LIN-42 Δ CK1BD to perform IP and check for binding of KIN-20.

As detailed in section 4, we cannot tag LIN-42a individually due to the structure of the genomic locus, and its level appear very low to begin with.

5. In the molting timing assay, there is an unexpected result where the delta-C-terminal-tail lin-42 allele resembles the n1089 (N-terminal deletion) (line 315). Could the authors more clearly explain this finding?

Revision Plan

As we point out in the manuscript, *n1089* is a partial deletion with a breakpoint in a noncoding (intronic) region of *lin-42*. Accordingly, it is currently unknown, what mature transcripts and proteins are made in the mutant animals. This prevents us from making educated guesses as to why there is a phenotypic resemblance between these and *lin-42Δtail* mutant animals. We will clarify in the manuscript that this is an interesting, but currently unexplained observation.

Minor comments:

1. The correspondence between the LIN-42 "SYQ" and "LT" motifs and the motifs referred to as "A" and "B" should be clarified, and consistent names/labels used. Are these interchangeable names? If it is necessary to use both names, the differences between SYQ/LT and A/B should be made more clear.

We agree that the situation is not completely satisfactory but feel that we need to use both names since they have both been used in the literature. We will work to revise the text to reflect more clearly the correspondence.

2. For data presented as "% of animals", please indicate the number of animals scored (e.g. egl, alae assays - ~ how many animals per replicate (dot)?).

We will provide these numbers.

3. Line 145-148 - Mentioning the relevant phenotype(s) of the *lin-42* null allele from the cited paper would provide a good point of comparison here.

We will mention the previously described phenotypes.

4. Line 201 - the phrase "This is also true for the proteins:" is unclear, as the previous sentence states that both *lin-42* and *kin-20* mRNAs oscillate, while the next sentence says that only LIN-42 protein oscillates.

We apologize for the confusion and will correct the text.

5. Line 231 - please explain the significance of the 'lower response signal' in the BLI assay for the CK1BD(no tail).

We will clarify that the lower response signal observed for the CK1BD compared to the CK1BD+Tail (residues 402-589; same construct used in Fig. 3B) reflects its smaller molecular weight, which reduces the overall mass contribution to the BLI sensor.

6. Fig. 2 - C/D - the genotype lane labels should I think indicate an N-terminal rather

We will fix this mistake.

7. Fig. 6, line 367 - *lin-42* is variably described as promoting increased KIN-20 'nuclear

Revision Plan

accumulation' or 'localization'. I think that 'accumulation' is more accurate, as it doesn't imply a specific mechanism for the difference (transport vs stabilization, etc.)

We will revise the manuscript accordingly.

8. Fig 6B - an overlay of the panels or another way of quantifying the colocalization would make this result more clear.

We will supply the requested overlay.

Reviewer #3 (Significance (Required)):

This work presents a major mechanistic and conceptual advance in our understanding of the role of lin-42/Period, a conserved key regulator of *C. elegans* development. Previously, it was not clear if the heterochronic and circadian functions of lin-42 were genetically separable, nor was it known how LIN-42 physically interacted with the CK1 homologue. This work addresses these questions using precise genome engineering and detailed phenotypic and biochemical approaches. The work also reveals the conservation of bi-directional/reciprocal regulation between lin-42 and kin-20. The main limitations of the study, which can potentially be addressed as outlined in the 'major points' above, are that evidence should be provided that lin-42 phosphorylation depends on kin-20 in vivo, and that the CK1BD mediates the interaction in vivo (since the in vitro work is with human CK1). As the authors indicate, this is the first 'conserved clock module' of this type, and this work will therefore be of significant interest to both the *C. elegans* developmental biology and the more general biological timing fields.

Field of expertise of the reviewer- *C. elegans* genetics and development.

3. Description of the revisions that have already been incorporated in the transferred manuscript

Please insert a point-by-point reply describing the revisions that were already carried out and included in the transferred manuscript. If no revisions have been carried out yet, please leave this section empty.

4. Description of analyses that authors prefer not to carry out

Please include a point-by-point response explaining why some of the requested data or additional analyses might not be necessary or cannot be provided within the scope of a revision. This can be due to time or resource limitations or in case of disagreement about the necessity of such additional data given the scope of the study. Please leave empty if not applicable.

@Ref. 2:

Revision Plan

Major comment 5 (Optional):

In this study, the authors carefully performed in vitro kinase assays, and I strongly suggest that they investigate whether the CKI-mediated phosphorylation of LIN-42 is temperature-compensated and whether the CKI-BD-AB regions affect it.

Temperature compensation is of course one of the most striking features of circadian clocks, and CK1-mediated phosphorylation of PER appears a critical component. We agree that it would be interesting to examine whether or not this feature exists in an animal whose development is not or only partially temperature-compensated. However, these studies are not straightforward – we would first have to set up an assay and demonstrate temperature compensation for the mammalian PER – CK1 pair as a positive control. We were not able to purify KIN-20 so could only test whether the LIN-42 substrate promoted temperature compensation. Moreover, either result for LIN-42 – CK1 would immediately raise new questions that would deserve extensive follow-up: if there is temperature compensation, why is worm development not compensated? If there is none, where/how do the interactions between CK1 and LIN-42 differ from those between CK1 and PER? Hence, we propose that these studies are outside the scope of the current study.

Major comment 6 (Optional):

In Figure 6, the authors argue that the CKI-BD of LIN-42 is important for CK1 nuclear translocation. It would be better to show the effect of the nuclear accumulation of CKI on nuclear proteins, like the mammalian CKI-PER2-CLOCK story. Does CKI localization affect phosphorylation status of other clock-related proteins including REV-ERB/NHR-85? Phospho-proteome analysis would identify nuclear substrates of CK1. In addition, is phosphorylation of LIN-42 dispensable for the CK1 nuclear translocation?

We agree that it will be important to identify relevant targets of KIN-20 in future work. Unfortunately, at this point, none are known, and we especially do not have any knowledge of the phosphorylation status of NHR-85. Indeed, in unrelated (and unpublished) work we have done a phosphoproteomics time course of wild-type animals. We have not detected any NHR-85-derived phosphopeptides in our analysis. Thus, this would establish a completely new line of research, incompatible with the timelines of a revision.

@Ref. 3:

1. The authors have shown that LIN-42 is phosphorylated in vivo, but the dependence of this phosphorylation on KIN-20 is not fully addressed. In the discussion (lines 417-420), the authors mention that the unhealthy phenotype of the kin-20 mutant animals prevented them from assessing LIN-42 phosphorylation in this genetic background. To bolster their model and to circumvent this issue, it should be feasible to generate a kin-20 degron allele and to perform the LIN-42 phospho-proteomics upon inducible degradation. Alternatively, perhaps a phos-tag western blot for LIN-42 could be used to compare the kin-20 wild-type to kin-20 mutants.

We agree, and acknowledged in the discussion, that phosphorylation of LIN-42 by KIN-20 in vivo has not been demonstrated by us. However, since our data from the LIN-42 Δ Tail

mutant also suggest that LIN-42 phosphorylation be functionally largely dispensable for KIN-20's function in rhythmic molting, we consider further elucidation of this point a lower priority, especially considering the challenges involved. As we have seen for our unpublished work on wild-type animals, a phosphoproteomics experiments would be costly and time-consuming, with a non-trivial analysis (due to the underlying dynamics of protein level changes). A phos-tag gel would be subject to multiple confounders given the abundance of the phosphosites that we detected on immunoprecipitated LIN-42 – unlikely to stem only from KIN-20 activity – and an increase in total LIN-42 levels that we observe upon KIN-20 depletion, and thus appears unsuited to providing a meaningful answer.

4. Related to points 2 and 3 above, the authors have shown that the CK1BD mediates association with human CK1 in vitro, and is required for nuclear accumulation of KIN-20 in vivo, but not that the complex formation between LIN-42 and KIN-20 depends on the CK1BD. Given the reciprocal co-IP findings, it should be feasible to create tagged versions of lin-42(Δ CK1BD) and to determine the effect on LIN-42-KIN-20 complex formation. While there is already a b-isoform tag, an a-isoform tag would also help to address whether both the b and a isoforms interact with KIN-20 in a CK1BD-dependent manner in vivo. These strains would also allow the authors to determine how the CK1BD deletion affects overall levels/stability/rhythmic accumulation of LIN-42(a or b), which would potentially serve to strengthen their conclusions about the role of the lin-42 CK1BD.

As detailed in section 2, we will address the point concerning LIN-42 Δ CK1BD. However, due to the overlapping exons, we are unable to tag the a-isoform independently of the b-isoform. Moreover, in a western blot of a line where both a- and b-isoforms are tagged, we have observed only little or no LIN-42a signal, suggesting that, like the c-isoform, its expression may be more limited, making biochemical characterization difficult. Hence, these experiments are not feasible.

Dear Dr. Ward,

Thank you for submitting your manuscript for consideration by the EMBO Journal. I have now read your manuscript, the reviewer comments and your response to them. Based on our editorial assessment and the referees' positive evaluations, I would like to invite you to submit a revised version of the manuscript along the lines indicated in your revision plan.

We generally allow three months as standard revision time. Should you foresee a problem in meeting this three-month deadline, please let us know in advance in order to arrange an extension. As a matter of policy, competing manuscripts published during this period will not negatively impact on our assessment of the conceptual advance presented by your study. However, please contact me as soon as possible upon publication of any related work to discuss the appropriate course of action.

When preparing your letter of response to the referees' comments, please bear in mind that this will form part of the Review Process File and will therefore be available online to the community. For more details on our Transparent Editorial Process, please visit our website: <https://www.embopress.org/page/journal/14602075/authorguide#transparentprocess>. Please also see the attached instructions for further guidelines on preparation of the revised manuscript.

Please feel free to contact me if you have any further questions regarding the revision. Thank you for the opportunity to consider your work for publication. I look forward to receiving your revised manuscript.

With best regards,

Ieva Gailite

Please remember: Digital image enhancement is acceptable practice, as long as it accurately represents the original data and

conforms to community standards. If a figure has been subjected to significant electronic manipulation, this must be noted in the figure legend or in the 'Materials and Methods' section. The editors reserve the right to request original versions of figures and the original images that were used to assemble the figure.

We realize that it is difficult to revise to a specific deadline. In the interest of protecting the conceptual advance provided by the work, we recommend a revision within 3 months (16th Jun 2025). Please discuss the revision progress ahead of this time with the editor if you require more time to complete the revisions.

Rev_Com_number: RC-2024-02838

New_manu_number: EMBOJ-2025-120783-T

Corr_author: Ward

Title: A conserved chronobiological complex times *C. elegans* development

Response to reviewers

Reviewer #1 (Evidence, reproducibility and clarity (Required)):

The authors investigate in this study the function of LIN-42 for the process of precise molting timing in *C. elegans*. To achieve this, they compare LIN-42 with its mammalian ortholog, Period. They found that similar to Period, LIN-42 interacted with the kinase KIN-20, a mammalian Casein kinase 1 (CK1) ortholog. Hence, two different proteins involved in rhythmic processes, LIN-42 and Period function in a conserved manner.

First, they used mutants with specific deletions to untangle various phenotypes during *C. elegans* development. From this analysis they identify a specific region, corresponding to a CK1-binding region in mammals, to be mainly involved in the rhythmic molting phenotype. Next, they identify KIN-20, the CK1 ortholog as interaction partner of LIN-42. They even were able to demonstrate an interaction of CK1 with the region of LIN-42. Using CK1, they identified potential phosphorylation sites within LIN-42 and compared those with immunoprecipitated protein in vivo. There was a substantial overlap. While the C-terminal tail of LIN-42 was heavily phosphorylated, deletion of the C-terminal part resulted only in a minor phenotype for rhythmic molting. Last but not least, they demonstrated that point mutations that inactivate the catalytic function of KIN-20 produced a rhythmic molting phenotype. The interaction of LIN-42 with KIN-20 affected the localization of the kinase, similar to what was found to Period and CK1.

Overall, the experiments are well done, well controlled and well described even for non-specialists. I guess it was not easy to kind of sort out the many overlapping phenotypes. It was certainly helpful just to focus on the clear rhythmic molting phenotype.

I have no major or minor comments.

Reviewer #1 (Significance (Required)):

The manuscript is well written and can be followed by non-specialists of the field. The experiments are well performed. Even if some experiments did not yield the expected phenotype, e.g. deletion of the C-terminal tail of LIN-42 had only a minor phenotype inspire of heavy phosphorylation, these experiments are anyhow included and explained.

Overall, the study is interesting for people in the *C. elegans* field and by similarity mammalian chronobiology. I would expect that most of the progress based on this study will be on the further elucidation of the molting phenotype and how the other phenotypes related to this. Then this could emerge as a blueprint for molting phenomena in other species as well.

I am a mammalian chronobiologist working on Period proteins.

We thank the reviewer for their positive evaluation of our work.

Reviewer #2 (Evidence, reproducibility and clarity (Required)):

This study represents pioneering work on LIN-42, the *C. elegans* ortholog of PER, uncovering its role in molting rhythms and heterochronic timing. A key strength of this work lies in its integrative approach, combining genetic and developmental analyses in *C. elegans* with biochemical characterization of LIN-42 protein.

At the organismal level, the authors take advantage of the power of *C. elegans* as a model system, employing precise genetic manipulations and high-resolution developmental assays to dissect the contributions of LIN-42 and its interaction partner KIN-20, the *C. elegans* ortholog of CK1, to molting rhythms. Their findings provide *in vivo* evidence that binding of LIN-42 with KIN-20 promotes the nuclear accumulation of KIN-20 and is crucial for molting rhythms, while its PAS domain appears dispensable for this function. This detailed phenotypic analysis of multiple LIN-42 and KIN-20 mutants represents a significant contribution to our understanding of the developmental clock.

At the biochemical level, the study provides a detailed analysis of the mechanism underlying LIN-42's interaction with CK1, demonstrating that LIN-42 contains a functionally conserved CK1-binding domain (CK1BD). Through their *in vitro* kinase assays and structural insights, the authors identified distinct roles for CK1BD-A and CK1BD-B: the former in kinase inhibition and the latter in stable CK1 binding and phosphorylation. Importantly, their data align well with previous findings on PER-CK1 regulation in mammalian and *Drosophila* systems, reinforcing the evolutionary conservation of key clock components.

Overall, this work stands out for its deep and important insights into how CK1-mediated regulation extends beyond the circadian clock to regulate the developmental clock. The combination of genetic approaches with biochemical analyses makes this an outstanding contribution to both chronobiology and nematode developmental biology.

We thank the reviewer for the strong endorsement of our work.

Major comment 1:

In Figure 2D, I could not find a crucial control if the authors claim that KIN-20 binds to LIN-42. For example, a single mutant of LIN-42-3xFLAG could be used as a control for the double mutant.

We have performed this control experiment and added it to Figure 2D.

Major comment 2:

The sizes of the KIN20 bands were very diverged (~40 kDa and ~60 kDa), but the authors provide no explanation for this.

We apologize for this omission. The worm produces several KIN-20 isoforms; we have added this information in the new Appendix Fig S2.

Major comment 3:

Regarding the MS study, the raw data are available, but the detailed supplemental Excel files would be more informative for readers. For example, are other interactors such as REV-ERB/NHR-85 detected in Figure 2A? Regarding Figure 4F, the list of phosphorylation sites and MS scores is also informative.

We apologize for our omission in stating clearly in the figure legend that the significantly enriched proteins were labeled with a red dot. This information is now in the legend. These were only LIN-42 itself and KIN-20. NHR-85 was not enriched. We now state this explicitly in the revised manuscript. We have also added the supplemental excel file Table S2.

Major comment 4:

It is an important finding that the PAS domain of LIN-42 is not essential for the molting rhythms. Is the PAS domain also dispensable for binding with KIN-20?

We have performed this experiment and found that PAS does not bind CK1. We have added this data as Appendix Fig S3.

Major comment 5 (Optional):

In this study, the authors carefully performed *in vitro* kinase assays, and I strongly suggest that they investigate whether the CKI-mediated phosphorylation of LIN-42 is temperature-compensated and whether the CKI-BD-AB regions affect it.

Although this is an interesting question, addressing it appears outside the scope of the manuscript and a revision. Temperature compensation is of course one of the most striking features of circadian clocks, and CK1-mediated phosphorylation of PER appears a critical component. We agree that it would be interesting to examine whether or not this feature exists in an animal whose development is not or only partially temperature-compensated. However, these studies are not straightforward – we would first have to set up an assay and demonstrate temperature compensation for the mammalian PER – CK1 pair as a positive control. We were not able to purify KIN-20 so could only test whether the LIN-42 substrate promoted temperature compensation. Moreover, either result for LIN-42 – CK1 would immediately raise new questions that would deserve extensive follow-up: if there is temperature compensation, why is worm development not compensated? If there is none, where/how do the interactions between CK1 and LIN-42 differ from those between CK1 and PER? Hence, we propose that these studies are outside the scope of the current study.

Major comment 6 (Optional):

In Figure 6, the authors argue that the CKI-BD of LIN-42 is important for CK1 nuclear translocation. It would be better to show the effect of the nuclear accumulation of CKI on nuclear proteins, like the mammalian CKI-PER2-CLOCK story. Does CKI localization affect phosphorylation status of other clock-related proteins including REV-ERB/NHR-85?

Phospho-proteome analysis would identify nuclear substrates of CK1. In addition, is phosphorylation of LIN-42 dispensable for the CK1 nuclear translocation?

This is another interesting question yet currently nothing is known about other CK1/KIN-20 targets, and we have no evidence for NHR-85 being one, nor do we have any information on its phosphorylation status *in vivo*. Indeed, in unrelated (and unpublished) work we have done a phosphoproteomics time course of wild-type animals. In our preliminary analysis, we have not detected any NHR-85-derived phosphopeptides. Thus, this would establish a completely new line of research, incompatible with the timelines of the revision.

To address whether phosphorylation of LIN-42 is dispensable for nuclear localization, we examined KIN-20 localization in *lin-42 ΔC-tail* mutant animals. We find that LIN-42ΔC-Tail mutants also have decreased nuclear accumulation of KIN-20. This is an intriguing result, because ΔCK1BD animals are much more arrhythmic than ΔC-tail animals. We infer that LIN-42 regulates KIN-20 by at least one additional mechanism beyond translocation/stabilization. Speculatively, based on our *in vitro* dissection of CK1 and LIN-42 interaction, we can propose that this mechanism involve regulation of KIN-20 enzymatic activity through binding to the CK1BD. We show these new results in the revised Figure 6 and consider their implications in the Discussion.

Major comment 7 (Optional):

LIN-42 rhythmic expression could drive rhythmic nuclear accumulation of KIN-20. It would be better to examine this possibility using kin-20::GFP in lin-42 mutants.

We agree that the mutant analysis is important for this and Fig. 6C shows reduced KIN-20 nuclear accumulation in LIN-42ΔCK1BD.

Minor 1:

I could not find the full gel images of the Western blot analyses as supplemental materials.

These data have been added to the source files.

Minor 2:

The authors discussed a conserved module in two different clocks. A statement regarding a recently published paper (Hiroki and Yoshitane, Commun Biol, 2024) would be informative for readers.

We have added such a statement.

****Referee cross-commenting****

I basically agree with reviewer 1 and hope that this paper will be published soon as it is very valuable for our field. I have constructively pointed out some parts that could be improved, but depending on the editor's judgement, I believe that even if not all of these are revised, it will be sufficient for publication.

Reviewer #2 (Significance (Required)):

This work stands out for its deep and important insights into how CK1-mediated regulation extends beyond the circadian clock to regulate the developmental clock. The combination of genetic approaches with biochemical analyses makes this an outstanding contribution to both chronobiology and nematode developmental biology.

I strongly suggest editors to accept this study with minor modifications according to the following comments.

We thank the reviewer for their strong support for publication and the clear indication of required vs. optional revisions.

Reviewer #3 (Evidence, reproducibility and clarity (Required)):

In their manuscript "A conserved chronobiological complex times *C. elegans* development", Spangler, Braun, Ashley et al. investigate the mechanisms through which the PERIOD orthologue, lin-42, regulates rhythmic molting in *C. elegans*. Through precise genetic manipulations, the authors identify a particular region of lin-42, the 'CK1BD', which regulates molting timing, with less effect on other lin-42 phenotypes (e.g. heterochrony). They show that LIN-42 and the casein kinase 1 (CK1) homologue KIN-20 interact in vivo, and identify phosphorylation sites of LIN-42. Using biochemical assays, they find that the CK1BD of LIN-42 is sufficient for interaction with the human homologue of KIN-20, CK1, in vitro. The LIN-42 CK1BD is also required for the proper nuclear accumulation of KIN-20 in vivo. Furthermore, a point mutation that should disrupt the catalytic activity of KIN-20 also shows an irregular molting phenotype, similar to the lin-42 CK1BD mutant.

The manuscript is very well-written and the data and methods are well-presented and detailed. Overall this work makes a convincing case that the *C. elegans* lin-42:Kin-20 and mammalian period:Ck1 interactions have functionally conserved roles in the oscillatory developmental programs of each organism (molting timing and circadian rhythms, respectively), with a few caveats below that can be addressed.

We thank the reviewer for their positive evaluation of our work.

Major comments:

1. The authors have shown that LIN-42 is phosphorylated in vivo, but the dependence of this phosphorylation on KIN-20 is not fully addressed. In the discussion (lines 417-420), the authors mention that the unhealthy phenotype of the kin-20 mutant animals prevented them from assessing LIN-42 phosphorylation in this genetic background. To bolster their model and to circumvent this issue, it should be feasible to generate a kin-20 degron allele and to perform the LIN-42 phospho-proteomics upon inducible degradation. Alternatively, perhaps a phos-tag western blot for LIN-42 could be used to compare the kin-20 wild-type to kin-20 mutants. We agree, and acknowledged in the discussion, that phosphorylation of LIN-42 by KIN-20 in vivo has not been demonstrated by us. However, since our data from the LIN-42 Δ Tail mutant also suggest that LIN-42 phosphorylation is less critical than binding for KIN-20's function in rhythmic molting, we consider further elucidation of this point a lower priority, especially considering the challenges involved. In particular, we consider a phosphoproteomics experiment outside the scope of this study, as we have learned from unpublished preliminary work on wild-type animals that these studies are costly and very time-consuming, and involve a non-trivial analysis (due to the underlying dynamics of protein level changes). A phos-tag gel would be subject to multiple confounders given the abundance of the phosphosites that we detected on immunoprecipitated LIN-42 – unlikely to stem only from KIN-20 activity – and an increase in total LIN-42 levels that we observe upon KIN-20 depletion. It thus appears unsuited to providing a meaningful answer.

2. For technical reasons, the in vitro biochemistry was done using human CK1 protein. There are a few places (e.g. results, line 248 and discussion line 437), where the language, in my opinion, is extrapolating the CK1 results too strongly to KIN-20. The authors mention that feedback inhibition is a known property of human CK1. It is indeed quite striking that the LIN-42 CK1BD region interacts with and is phosphorylated by the human counterpart of KIN-20, and that feedback inhibition is also seen! However, the language about KIN-20 itself should be softened, since there does not appear to be clear evidence that KIN-20 exhibits the same properties as human CK1 (unless perhaps human CK1 can functionally replace KIN-20 in worms, or the proteins were extremely similar?)

We have carefully examined the text for instances where we extrapolated too much and toned these down. (We note that this does not apply to the example of line 248 where we wrote “Collectively, our data establish that the LIN-42 CK1BD is functionally conserved and mediates stable binding to the CK1 kinase domain.”, i.e., there was no mentioning of KIN-20.) We do note that the kinase domains (which mediate the interaction) are indeed unusually similar, with overall 79% sequence identity - and 100% in the active site.

3. The role of the three LIN-42 isoforms should be further clarified. Minimally, it should be explained why the alleles where both b and c isoforms should be flag-tagged seem to only produce detectable b isoform (e.g. Fig. 2C).

We apologize for our incomplete figure labeling. Since both the *lin-42* and the *kin-20* locus are highly complex, giving rise to several isoforms, we were unsure whether the additional bands seen in Fig.2 and not originally labeled by us derived from LIN-42 and KIN-20. We now performed an additional experiment using a line containing only tagged LIN-42, which confirms that these are indeed additional LIN-42 isoforms (new Appendix Figure S2). Moreover, although available literature focuses on only three major isoforms, *lin-42a-c*, Wormbase lists several more, and we see evidence for these. Since we do not feel sufficiently confident, based on estimated size from the Western blot alone, to assign specific isoform identities, we now label the corresponding bands only as LIN-42, without further indication of isoform. We explain the situation in the main text and have also renamed the relevant strain to *3xflag::lin-42* (i.e., without isoform information but clarifying the position of the tag, which is also indicated in the schematic figures).

4. Related to points 2 and 3 above, the authors have shown that the CK1BD mediates association with human CK1 in vitro, and is required for nuclear accumulation of KIN-20 in vivo, but not that the complex formation between LIN-42 and KIN-20 depends on the CK1BD. Given the reciprocal co-IP findings, it should be feasible to create tagged versions of *lin-42(deltaCK1BD)* and to determine the effect on LIN-42-KIN-20 complex formation. While there is already a b-isoform tag, an a-isoform tag would also help to address whether both the b and a isoforms interact with KIN-20 in a CK1BD-dependent manner in vivo. These strains would also allow the authors to determine how the CK1BD deletion affects overall levels/stability/rhythmic accumulation of LIN-42(a or b), which would potentially serve to strengthen their conclusions about the role of the *lin-42* CK1BD.

We generated by genome-editing a FLAG-tagged *LIN-42ΔCK1BD* and attempted to perform IP and check for binding of KIN-20. Unfortunately, these animals grow extremely poorly on plate, and the one experiment for which we were able to collect enough material failed for technical reasons, i.e., we did not achieve a good enrichment of either LIN-42 or the SART-3 control bait. We cannot tag LIN-42a individually due to the structure of the genomic locus, and its level appears very low to begin with.

5. In the molting timing assay, there is an unexpected result where the delta-C-terminal-tail *lin-42* allele resembles the *n1089* (N-terminal deletion) (line 315). Could the authors more clearly explain this finding?

As we point out in the manuscript, *n1089* is a partial deletion with a breakpoint in a noncoding (intronic) region of *lin-42*. Accordingly, it is currently unknown what mature transcripts and proteins are made in the mutant animals. This prevents us from making educated guesses as to why there is a phenotypic resemblance between these and *lin-42Δtail* mutant animals. We clarify in the manuscript that this is an interesting, but currently unexplained observation.

Minor comments:

1. The correspondence between the LIN-42 "SYQ" and "LT" motifs and the motifs referred to as "A" and "B" should be clarified, and consistent names/labels used. Are these interchangeable names? If it is necessary to use both names, the differences between SYQ/LT and A/B should be made more clear.

We agree that the situation is not completely satisfactory but feel that we need to use both names since they have both been used in the literature. We revised the text in the abstract, introduction, and start of the results to make it clear that SYQ=CK1BD-A and LT=CK1BD-B.

2. For data presented as "% of animals", please indicate the number of animals scored (e.g. egl, alae assays - ~ how many animals per replicate (dot)?).

We added these numbers to the appropriate figure legends.

3. Line 145-148 - Mentioning the relevant phenotype(s) of the lin-42 null allele from the cited paper would provide a good point of comparison here.

We now mention the previously described phenotypes of the lin-42 null.

4. Line 201 - the phrase "This is also true for the proteins:" is unclear, as the previous sentence states that both lin-42 and kin-20 mRNAs oscillate, while the next sentence says that only LIN-42 protein oscillates.

We apologize for the confusion and have corrected the text.

5. Line 231 - please explain the significance of the 'lower response signal' in the BLI assay for the CK1BD(no tail).

We have clarified that the lower response signal observed for the CK1BD compared to the CK1BD+Tail (residues 402-589; same construct used in Fig. 3B) reflects its smaller molecular weight, which reduces the overall mass contribution to the BLI sensor.

6. Fig. 2 - C/D - the genotype lane labels should I think indicate an N-terminal rather

Thank you for pointing this out; we have fixed this mistake.

7. Fig. 6, line 367 - lin-42 is variably described as promoting increased KIN-20 'nuclear accumulation' or 'localization'. I think that 'accumulation' is more accurate, as it doesn't imply a specific mechanism for the difference (transport vs stabilization, etc.)

We have revised the manuscript accordingly.

8. Fig 6B - an overlay of the panels or another way of quantifying the colocalization would make this result more clear.

We have provided the requested overlay.

Reviewer #3 (Significance (Required)):

This work presents a major mechanistic and conceptual advance in our understanding of the role of lin-42/Period, a conserved key regulator of *C. elegans* development. Previously, it was not clear if the heterochronic and circadian functions of lin-42 were genetically separable, nor was it known how LIN-42 physically interacted with the CK1 homologue. This work addresses these questions using precise genome engineering and detailed phenotypic and biochemical approaches. The work also reveals the conservation of bi-directional/reciprocal regulation between lin-42 and kin-20. The main limitations of the study, which can potentially be addressed as outlined in the 'major points' above, are that evidence should be provided that lin-42 phosphorylation depends on kin-20 in vivo, and that the CK1BD mediates the interaction in vivo (since the in vitro work is with human CK1). As the authors indicate, this is the first 'conserved clock module' of this type, and this work will therefore be of significant interest to both the *C. elegans* developmental biology and the more general biological timing fields.

Field of expertise of the reviewer- *C. elegans* genetics and development.

Dear Jordan,

Thank you for submitting a revised version of your manuscript. We have now received input from all original reviewers, who are satisfied with the revisions and now recommend acceptance of the manuscript. There now remain only a few editorial points that need to be addressed before I can extend official acceptance of the manuscript:

1. Please submit up to five keywords.
2. Confirmation email to the author Andrea Ramos Coronado was returned to the sender - please provide the current email address.
3. Please submit a complete author checklist, which you can download from our author guidelines (<https://www.embopress.org/pb-assets/embo-site/EMBO%20Press%20Author%20Checklist-1642513524327.xlsx>). Please insert information in the checklist that is also reflected in the manuscript. The completed author checklist will also be part of the Review Process File.
4. Please make sure that the order and headings of the sections are as follows: abstract, keywords, introduction, results, discussion, methods, acknowledgements, disclosure and competing interests statement, references, figure legends, expanded view figure legends.
5. CRedit has replaced the traditional author contributions section because it offers a systematic, machine-readable author contributions format that allows for more effective research assessment. Please remove the Authors Contributions from the manuscript and use the free text boxes beneath each contributing author's name in our online submission system to add specific details on the author's contribution. More information is available in our guide to authors.
6. All Materials and Methods need to be described in the main text using our 'Structured Methods' format. According to this format, the Methods section includes a Reagents and Tools Table (listing key reagents, experimental models, software and relevant equipment and including their sources and relevant identifiers) followed by a Methods and Protocols section describing the methods, ideally using a step-by-step protocol format. The aim is to facilitate adoption of the methodologies across labs. Please download and fill our Reagents and Tools Table template (.docx), which you can find in our author guidelines: <https://www.embopress.org/page/journal/14602075/authorguide#structuredmethods>. The information currently included in Appendix Tables S3, S4 and S5 should be added in the appropriate sections of the Reagents and Tools table template as indicated. When submitting your revised manuscript, please do not include the Reagents and Tools Table in the Methods section of the manuscript but upload it as a separate file choosing the file type "Reagent Table". An example of a Method paper with Structured Methods can be found here: <https://www.embopress.org/doi/10.15252/msb.20178071>.
7. Please rename Appendix Table S2 into Table EV1. Please remove the legend from the manuscript text and add it directly to the table file.
8. Please compile the Appendix Figures and Appendix Table S1 in a single PDF, add a table of contents with page numbers, as well as the legends for appendix figures and tables, which should be removed from the manuscript text and placed underneath the corresponding table/figure.
9. Please upload source data as one (Zip) file per figure. There is an excel file labeled "Guin Numerical Source Data_figure 1_figure 5.xlsx"; please add this to the ZIP file for Figure 1 and for Figure 5.
10. Our data editors have flagged the following issues in figure legends that need correcting:
 - Please define the annotated p values ****/****/**/* as well as provide the exact p-values for the same in the legend of figure EV1 A-C; EV3 A-C as appropriate.
 - Please provide the exact p values in the legends of figures 1C-F; 3C, 4C, E; 5B-E; 6E.
 - Please indicate the statistical test used for data analysis in the legends of figures EV1 A-C; EV3 A-C.
 - Please define the box plots in terms of minima, maxima, centre, bounds of box and whiskers, and percentile in the legends of figures EV1 A-C; EV3 A-C.
 - Please define the measure of center for the error bars in the legends of figures 1C-F.

With kind regards,

leva

leva Gailite, PhD

Senior Scientific Editor
The EMBO Journal
Meyerhofstrasse 1
D-69117 Heidelberg
Tel: +4962218891309
i.gailite@embojournal.org

Revision to The EMBO Journal should be submitted online within 90 days, unless an extension has been requested and approved by the editor; please click on the link below to submit the revision online before 25th Nov 2025:

Referee #1:

I don't have any additional comments.

Referee #2:

No additional comments, as the authors have successfully addressed the comments from the previous review in their carefully revised manuscript.

As indicated in the previous reviews, this excellent study presents a major mechanistic and conceptual advance on the role of lin-42/Period and kin-20 in developmental timing. Understanding this conserved clock module will be of significant interest to both the *C. elegans* developmental biology and the broad biological timing fields.

Referee #3:

The authors have responded to my comments sincerely, and I consider the manuscript suitable for acceptance.

Rev_Com_number: RC-2024-02838

New_manu_number: EMBOJ-2025-120783R

Corr_author: Ward

Title: A conserved chronobiological complex times *C. elegans* development

The authors addressed the remaining editorial issues.

Dear Jordan,

Thank you for addressing the final formatting requests in your revised manuscript. I am now pleased to inform you that your manuscript has been accepted for publication in the EMBO Journal. Congratulations on a great study!

If you have any questions, please do not hesitate to contact the Editorial Office. Thank you for your contribution to The EMBO Journal!

With best wishes,

Ieva

Rev_Com_number: RC-2024-02838
New_manu_number: EMBOJ-2025-120783R1
Corr_author: Ward
Title: A conserved chronobiological complex times C. elegans development